# Co- and postseismic subaquatic evidence for prehistoric fault activity near Coyhaique, Aysén Region, Chile

Morgan Vervoort[1], Katleen Wils[1], Kris Vanneste[2], Roberto Urrutia[3], Mario Pino[4], Catherine Kissel[5], Marc De Batist[1], Maarten Van Daele[1]

[1]Renard Centre of Marine Geology (RCMG), Department of Geology, Ghent University, 9000 Ghent, Belgium
[2]Royal Observatory of Belgium, 1180 Brussels, Belgium
[3]Centro EULA, Universidad de Concepción, Casilla 160-C, Concepción, Chile
[4]Instituto de Ciencias de la Tierra, Universidad Austral de Chile, Casilla 567, Valdivia, Chile
[5]Laboratoire des Sciences du Climat et de l'Environnement/IPSL, CEA/CNRS/UVSQ, Université Paris-Saclay, 91198 Gif-sur-Yvette, France

*Correspondence to*: Morgan Vervoort (Morgan.Vervoort@UGent.be)

**Abstract.** Chilean Patagonia is confronted with several geohazards due to its tectonic setting, i.e., the presence of a subduction zone and numerous fault zones, e.g. the Liquiñe-Ofqui Fault Zone (LOFZ). This region has therefore been the subject of numerous paleoseismological studies. However, this study reveals that the seismic hazard is not limited to these large tectonic structures by identifying past fault activity near Coyhaique in the Aysén Region. Mass wasting deposits in Lago Pollux, a lake located ca. 15 km SW of this region's capital, were identified through analysis of reflection-seismic data and was linked to a simultaneous event recorded in nearby Lago Castor. Furthermore, a coeval ~50 year-long catchment response was identified in Aysén Fjord based on the multiproxy analysis of a portion of a sediment core. Assuming that this widely recognized event was triggered by an earthquake, ground-motion modelling was applied to derive the most likely magnitude and source fault. The model showed that an earthquake rupture along a local fault, in the vicinity of Lago Pollux and Lago Castor, with a magnitude of 5.6-6.8, is the most likely scenario.

## 1    Introduction

Lake and fjord sediments represent continuous archives of natural hazards, such as earthquakes or volcanic eruptions, in a variety of tectonic settings (e.g., Moernaut, 2020; Sabatier et al., 2022; St.-Onge et al., 2004). Seismic shaking in such basins can cause widespread and coeval mass wasting and/or surficial sediment remobilization, recorded as mass transport deposits (MTDs) and/or turbidites (e.g., Adams, 1990; Howarth et al., 2014; Kremer et al., 2017). The presence of such coeval deposits forms the basis of the synchronicity criterion (Adams, 1990; Schnellmann et al., 2002), which is typically invoked to infer an earthquake as most likely source-mechanism. Numerous studies have successfully applied this approach for the reconstruction of a regional seismic history (e.g., Beck et al., 2015; Gastineau et al., 2021; Kremer et al., 2017; Moernaut et al., 2007, 2014; Praet et al., 2017; Schnellmann et al., 2002; Strasser et al., 2007; Wilhelm et al., 2016; Wils et al., 2018, 2020). Apart from the coseismic effects of earthquake shaking, a long-term increased sediment yield may also occur in lacustrine or fjord basins due to earthquake-induced landslides and perturbations in their catchment. Several recent studies evidenced such postseismic enhanced sediment flux (e.g., Avşar et al., 2014; Howarth et al., 2012, 2014). Lake and fjord sediments may thus preserve both coseismic and postseismic imprints and are therefore valuable archives of past seismic activity.

Numerous paleoseismological studies have been conducted in Chile due to its active tectonic setting (e.g., Moernaut et al., 2007, 2014; Piret et al., 2018; Van Daele et al., 2015, 2019; Wils et al., 2018, 2020). In a recent study of the sedimentary infill of Lago Castor, located in the Andean mountains (45.6°S, 71.8°W), Van Daele et al. (2016) identified a turbidite (dated between 4300 and 4450 cal yrs BP) associated to multiple MTDs, suggesting it was triggered by an earthquake (synchronicity criterion), although the causative fault remains unclear. The subduction zone is likely too far away to be able to cause significant shaking in the area, as suggested by the fact that no damage was reported following the largest ever instrumentally

recorded 1960 $M_w$ 9.5 Valdivia earthquake (Lazo, 2008). A more established source of seismic hazard in the region is the Liquiñe-Ofqui Fault Zone (LOFZ) (Hervé, 1976), although the closest branch of this fault system is located about 50 km east of the lake. Some small, local fault traces have been identified closer to the lake, but their activity is largely unknown (De La Cruz et al., 2003; SERNAGEOMIN, 2003). In any case, the possibility of an earthquake occurring and generating strong seismic shaking close to Coyhaique, the district capital, is important for the hazard and risk assessment of the area, thus justifying further research into this past event.

In this respect, we studied the seismic stratigraphy of Lago Pollux (about 3 km southwest of Lago Castor) in order to reveal its sedimentary history and achieve a correlation with the sedimentary infill of Lago Castor. Identification of the presence or absence of synchronous large-scale mass-wasting in this nearby lake would significantly improve our understanding of the size and impact of this presumed paleo-earthquake. We also analyzed a section of a sediment core retrieved from inner Aysén Fjord, located at the same latitude but closer to the megathrust and on the LOFZ, for comparison with the Lago Pollux and Lago Castor records. In Aysén Fjord, multiple sedimentary imprints of both megathrust and crustal earthquakes along the LOFZ were already identified (Wils et al., 2018; 2020). By combining the sedimentary data in these two lakes and one fjord with ground-motion modelling, we aim to identify the most likely location and magnitude of this ~4400 cal yrs BP paleo-earthquake.

## 2  Study area

### 2.1  Regional setting

Lago Pollux is located in southern Chile, in the Aysén Region on the eastern side of the Patagonian Andes, close to the Argentinian border (Figure 1a). Chilean Patagonia has a humid climate with low seasonality and a west-to-east precipitation gradient resulting from the dominant Southern Westerly Winds (SWW or southern westerlies) (Gilli et al., 2005; Henry, 2005). The vegetation surrounding the lake is mainly pastureland, covered by fallen and burnt logs resulting from deforestation of the subantarctic *Nothofagus pumilio* forest (Markgraf et al., 2007). Lago Pollux has a surface area of 9.06 km$^2$ and is situated ~680 m above sea level (m asl), on a plateau where also Lago Castor and Lago Thompson are located (Figure 1b). Their catchment mainly contains Cretaceous sedimentary rocks and Cenozoïc volcanoclastics (De La Cruz et al., 2003; SERNAGEOMIN, 2003), and the surrounding hills reach heights of over 1000 m asl. The main inflow for Lago Pollux comes from Lago Castor and Lago Thompson, which are situated around 20 m and 70 m higher, respectively. These two lakes have no major inflow apart from some surrounding creeks. Lago Pollux then drains towards Lago Frío, located about 150 m lower at the base of the plateau, via a small river at its central western side. Finally, Lago Frío drains via the Río Simpson, eventually flowing through Río Aysén into Aysén Fjord, located on the western side of the Patagonian Andes (Figure 1a, c). The geology in the area surrounding Aysén Fjord is dominated by igneous rocks such as granites, (grano-)diorites and tonalites of the North Patagonian Batholith. However, an important part of the catchment and that of the feeding rivers (i.e., Río Cuervo, Río Condor, Río Mañihuales and Río Simpson) is composed of basaltic to rhyolitic Quaternary volcanic centers (De La Cruz et al., 2003; SERNAGEOMIN, 2003).

### 2.2  Volcanotectonic setting

The Aysén Region is seismically active, owing to its location close to the Peru-Chile subduction trench, where the oceanic Nazca Plate obliquely subducts below a continental portion of the South American Plate. The oblique subduction additionally results in the presence of a large crustal strike-slip fault system, the Liquiñe-Ofqui Fault Zone (LOFZ; Figure 1a), of which one of the major fault strands crosses Aysén Fjord (Legrand et al., 2011; Métois et al., 2012). Between about 44 and 46°S, east of the main LOFZ fault strands and Aysén Fjord, two major north-northeast trending faults can be distinguished: the Azul Tigre and Río Mañihuales Faults (Figure 1a) (Thomson, 2002). However, none of these major tectonic structures reach towards

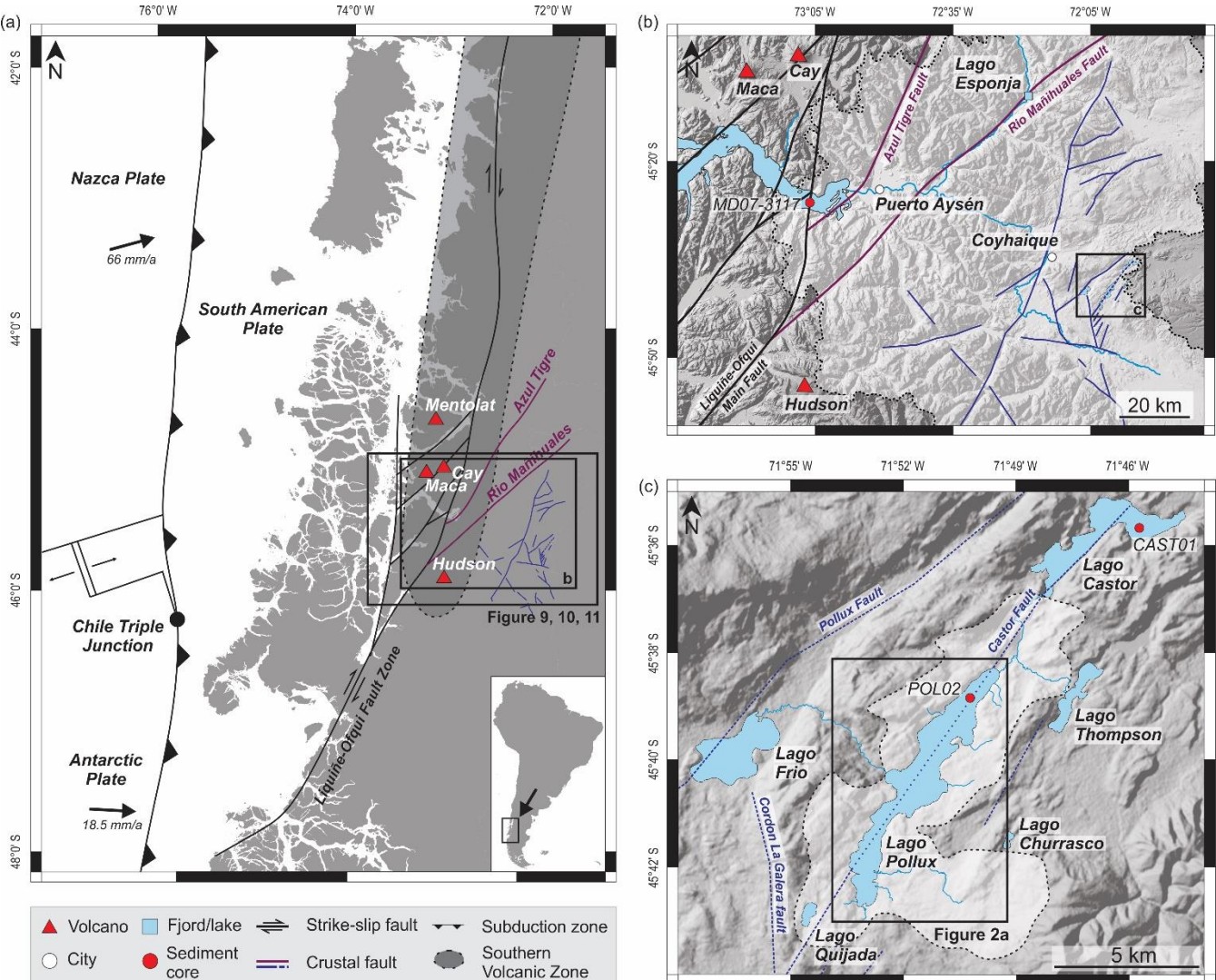

**Figure 1. (a) Structural geological map of Chile between 42° and 48° S. The Chilean subduction zone and plate motions (convergence rates after Wang et al., 2007) are indicated (arrows), as well as the Liquine-Ofqui Fault Zone (LOFZ) and smaller faults surrounding the city of Coyhaique (De La Cruz et al., 2003). (b) Shaded NASA Shuttle Radar Topography Mission Global 1 arc (SRTMGL1N) terrain data (NASA JPL, 2013) of the study area where the inner Aysén Fjord is indicated, as well as its catchment area (light shaded zone) and major rivers. The LOFZ and local fault traces are again indicated, as well as the location of the retrieved sediment core MD07-3117. (c) SRTMGL1N terrain data (NASA JPL, 2013) of the area surrounding Lago Pollux, including Lago Castor, Lago Frio, Lago Thompson, Lago Churrasco, Lago Quijada and local faults. The catchment area of Lago Pollux (light shaded zone) and major rivers surrounding Lago Pollux (light blue), as well as the location of short sediment cores in both Lago Pollux and Lago castor are indicated.**

the area of Lago Castor and Lago Pollux, where only multiple local smaller crustal strike-slip faults have been mapped (Figure 1a, b; De La Cruz et al., 2003; SERNAGEOMIN, 2003). Apart from seismic activity, the presence of a subduction zone also results in an active volcanic arc. From ~33°S to 46°S, the volcanic arc is referred to as the Southern Volcanic Zone (SVZ), characterized by complex volcano-tectonic interactions (Cembrano and Lara, 2009). An illustration of this interaction is the

2007 seismic swarm affecting Aysén Fjord (Agurto et al., 2012; Legrand et al., 2011). At first, the swarm was thought to have a solely tectonic origin (Cembrano et al., 2007), but it was later confirmed to have a fluid-driven source mechanism as well (Legrand et al., 2011). In this case, both magma and water, from Aysén Fjord itself or hydrothermal vents therein, may have interacted to generate the seismic swarm. Some of the active volcanoes in the Aysén Region are the Mentolat, Cay, Macá and Hudson volcanoes (Figure 1a) (Naranjo and Stern, 2004; Stern, 2004). The Hudson Volcano is the southernmost and most

active volcano of the SVZ (Naranjo and Stern, 2004; Weller et al., 2018). Due to the prevailing winds, most of its eruptions had ashes dispersed in an eastward direction (Naranjo and Stern, 1998), towards the Lago Castor and Pollux area, where the resulting tephra deposits can thus serve as regional markers. The three largest Holocene eruptions of the Hudson Volcano are H1, H2 and H3 with ages of 8.50-8.01 cal yr BP, 4.09-3.61 cal yr BP and 1991 AD, respectively (Naranjo and Stern, 1998, 2004; Weller et al., 2014), all of which have been identified in the sedimentary infill of Aysén Fjord (Wils et al., 2018; 2020).

An even larger eruption, with an estimated tephra volume of more than 20 km$^3$, has been described by Weller et al. (2014). This event, called Ho, has a pre-Holocene age of 17.30-17.44 cal yr BP and its tephra has been identified in Lago Churrasco and Lago Quijada, located less than 3 km east and 1.5 km south-west of Lago Pollux, respectively (Figure 1a, b). In addition to the Ho tephra, also the H2 tephra has been identified in both of these lakes (Moreno et al., 2019). Also in Lago Castor, both the Ho and H2 deposits were identified, alongside an additional 52 tephra deposits (Van Daele et al., 2016).

## 3    Material and methods

### 3.1    Lago Pollux and Lago Castor

A geophysical survey with both a CENTIPEDE sparker and GEOPULSE pinger source system, from the Renard Centre of Marine Geology (RCMG – Ghent University), resulting in high resolution sub-bottom seismic profiles, was conducted in December 2009 on Lago Pollux (Figure S1). The sparker produces a broad-spectrum seismic signal (0.4-1.5 kHz), with a mean
frequency of ~1.3 kHz, resulting in a maximum vertical resolution of ~0.5 m. A single-channel streamer with 10 hydrophones was used as a receiver. The pinger operates at a mean frequency of ~3.5 kHz, resulting in a slightly higher vertical resolution of ~0.2 m. The survey was conducted in conjunction with a similar geophysical survey on neighboring Lago Castor. In addition, short sediment cores (~60 cm) were taken in both lakes (Figure 1b), as well as a long sediment core (>15 m) in Lago Castor. The short gravity cores were split, imaged and their magnetic susceptibility (MS) was measured with a Bartington MS2E point
sensor at a 2.5 mm step size. These cores were already described and the results interpreted. The long Lago Castor core has a composite length of 15.4 m and was described and interpreted by Van Daele et al. (2016). An age model of the composite core was achieved based on 20 radiocarbon ages from 13 bulk organic matter and 7 terrestrial macroremains samples. These samples were measured with the Accelerator Mass Spectrometry (AMS) at NOSAMS (MA, USA) and calibrated using the calibration curve for the Southern Hemisphere (SHCal13; Hogg et al., 2013). A final age model was constructed with CLAM (version
2.2; Blaauw, 2010) (Figure S2, Van Daele et al., 2016).The sedimentary history of Lago Pollux is presumed to be similar to that of Lago Castor due to their proximity and hydrographic connection (Figure 1b). The seismic data of Lago Pollux was interpreted using the S&P Global Software Geoscience Package (Kingdom version 2019), allowing for identification of several seismic units of which the thickness was estimated based on the corresponding acoustic velocities from Lago Castor (Van Daele et al., 2016). A bathymetric map was produced by gridding of the lake floor reflector picks using Surfer (Golden
Software). An acoustic velocity of 1450 m/s was assumed for water of 10 °C (Del Grosso, 1974), which is about the average water temperature in Lago Pollux, as determined by a CTD profile at the time of the seismic survey (Van Daele et al., 2016). Due to the lower vertical resolution, the sparker profiles were only used for the reconstruction of the bathymetric map and thickness estimation of the units, while the pinger profiles were further used for geophysical data analysis.

### 3.2    Aysén Fjord

A Calypso sediment core, MD07-3117, of 21.14 m length was retrieved from inner Aysén Fjord during the PACHIDERME cruise on board of the RV *Marion Dufresne* in February 2007. Wils et al. (2018, 2020) studied the core in detail to create an event record of earthquakes and volcanic eruptions for the Aysén Fjord region. For that purpose, the archive half was scanned with the Ghent University Hospital medical X-ray computed tomography (CT) scanner (Siemens; SOMATOM Definition Flash; Siemens AG) at ~0.2 mm resolution in x and y directions, and a 0.6 mm z-resolution. VGStudio 3.2 was used for
visualization. Radiocarbon dating was performed on 23 terrestrial macroremains. The samples were analyzed using the ARTEMIS AMS facility in Saclay (Moreau et al., 2013). A chronology was achieved by classical age-depth modelling using CLAM (version 2.2; Blaauw, 2010) and the SHcal13 calibration curve (Hogg et al., 2013). The present study focusses on sections VIII and IX of the core (9.5-12 m depth) (Figure S3, Kissel et al., 2007), which has an age range (Wils et al., 2020) covering that of the mass-wasting event identified in the sedimentary infill of Lago Castor (Van Daele et al., 2016). Both

sections were logged with the Geotek Multi Sensor Core Logger (MSCL) of Ghent University to obtain high-resolution spectrophotometric (Konica Minolta CM-2600d) and MS data at a 2 mm interval. The acquired color reflectance data was used to calculate the normalized relative absorption band area between 400 and 560 nm ($nRABA_{400-560}$) and reflectance ratio between 590 and 690 nm ($R_{590}/R_{690}$). The former has been used as a proxy for total organic carbon (TOC) content (Vandekerkhove et al., 2020) and is calculated by the following formula (Rein and Sirocko, 2002):

$$nRABA_{400-560} = \left[ \left( \frac{R_{590}}{R_{400}} \right) + \cdots + \left( \frac{R_{590}}{R_{560}} \right) \right] / R_{mean}$$

in which $R_{590}$ is the reflectance at 590 nm, $R_{400}$ at 400 nm and $R_{560}$ at 560 nm. $R_{mean}$ is the average reflectance of all measured reflectance values (360 – 740 nm). The $R_{590}/R_{690}$ index has been used as a proxy for the presence of illite, biotite and chlorite and is calculated by dividing the reflectance value at 590 nm ($R_{590}$) by the reflectance value at 690 nm ($R_{690}$) (Trachsel et al., 2010).

Two sample sets, each with a total of 35 samples (taken at a 1 cm interval), were taken in section IX of core MD07-3117 for grain-size analysis and estimation of organic matter content. This includes four samples in a previously-identified turbidite that has been related to a megathrust earthquake (Wils et al., 2020). For each sample, grain-size analysis was performed with a Malvern Mastersizer 3000 after chemical pre-treatment to remove organic matter, calcium carbonate and biogenic silica (procedure after Van Daele et al., 2016). The grain-size data was subsequently unmixed into different grain-size populations or end members (EM), using the MATLAB-based toolbox AnalySize (Paterson and Heslop, 2015). Additionally, loss-on-ignition (LOI) was performed to estimate the organic matter content (Heiri et al., 2001). The samples were first dried for 24 hours at 105°C ($LOI_{105}$), after which they were heated to 550°C for four hours ($LOI_{550}$). The $LOI_{550}$ values were calculated using the dry sediment weight ($LOI_{105}$).

Ten samples were selected to determine the elemental (C and N) and stable isotope ($\delta^{13}C$) composition in order to identify the source – terrestrial or marine – of the sedimentary organic matter (Carneiro et al., 2021). The optimal sample weight was calculated based on the $LOI_{550}$ values. The samples were placed in silver capsules and treated with 25 μl of sulfurous acid ($H_2SO_3$, 6-8%) to remove any inorganic carbon. The organic geochemical analysis was carried out by the Isotope Bioscience Laboratory (ISOFYS) of Ghent University with the Elemental Analyzer – Isotope Ratio Mass Spectrometer-I (EA-IRMS-I ANCA-GSL), interfaced with a 20-22 IRMS, SerCon. The standard deviation on the $\delta^{13}C$ measurements is < 0.6 ‰ on the Vienna Pee Dee Belemnite (VPDB) scale. Normalization on VPDB scale was done using two soil references with certified $\delta^{13}C$ values (-22.69 ± 0.04 ‰ and -28.85 ± 0.1 ‰).

### 3.3 Ground-motion modelling

Our analysis of the sedimentary infill of Lago Pollux and Aysén Fjord in the period around 4400 cal yrs BP, allow us to identify the presence or absence of sedimentary deposits that can be related to the major mass-wasting event identified in Lago Castor. For coeval deposits over a large area, an earthquake-related triggering mechanism is generally assumed. In that case, the presence or absence of coeval deposits can be used as positive or negative evidence for seismic shaking, respectively. To generate coseismic deposits (positive evidence), a minimum level of shaking is required, depending on the type of deposits. Ideally, threshold values are determined based on local calibration using positive and negative evidence of historical earthquakes, e.g. Moernaut et al. (2014); Van Daele et al. (2015); Wilhelm et al. (2016). However, especially for the Lago Castor area, no historical events are available to conduct such site-specific calibration. Therefore, we use average values based on evaluation of global thresholds (e.g., Van Daele et al., 2020; Vanneste et al., 2018). For major onshore landslides and rockfalls, a shaking intensity of VII½ or higher is generally required (e.g., Keefer, 1984; Serva et al., 2016), and is confirmed by the observations following the 2007 Mw 6.2 earthquake in Aysén Fjord (Naranjo et al., 2009; Sepúlveda et al., 2010). Subaquatic mass-movements can occur with intensities of VI½ and higher (e.g., Moernaut et al., 2014; Monecke et al., 2004; Van Daele et al., 2015; Wilhelm et al., 2016). Delta failures, with an external triggering mechanism, require the least intense

shaking, and can occur when intensity V½ is exceeded (e.g., Moernaut et al., 2014; Van Daele et al., 2020). The absence of such deposits (negative evidence) implies that these threshold levels of ground shaking were probably not exceeded. To accommodate for variability in preconditioning factors, such as slope angle or availability of sediment (e.g., Bernhardt et al., 2015; Molenaar et al., 2019; Strasser et al., 2007) that may affect the susceptibility to slope failure, we added ½ intensity unit for the threshold value associated to negative evidence. As smaller increments constrain too much and larger increments would imply that negative evidence has no effect on the outcome, the ½ intensity increment is a good compromise (Kremer et al., 2017; Vanneste et al., 2018). In other words, the absence of onshore landslides implies that intensities of VIII were not reached, as exceeding this value is assumed to always result in onshore mass wasting.

The most likely source location and magnitude for the considered earthquake can be estimated using the observed set of positive and negative earthquake evidence, in combination with their respective intensity thresholds, and by applying the probabilistic ground-motion modelling methodology developed by Vanneste et al. (2018) using the open-sourced engine OpenQuake (Pagani et al., 2014). This approach uses an intensity prediction equation (IPE) to calculate the shaking intensity for any site in the vicinity of an earthquake with known magnitude and location. By calculating the intensities caused by a range of hypothetical earthquakes with different magnitudes and locations in an arbitrary grid that comprises the study sites, they can be compared to the intensity thresholds found at Lago Pollux, Lago Castor and Aysén Fjord. The probability for each of these potential earthquakes to have caused the observed pattern of positive and negative evidence can thus be calculated. For this study, the IPE developed by Bakun and Wentworth (1997) was chosen, as it has proven to adequately represent the seismic shaking observed during crustal earthquakes in the Aysén Region (Vanneste et al., 2018). However, as the BakunWentworth1997 IPE does not have an explicit uncertainty term needed for probabilistic calculations, we arbitrarily considered a 1σ standard deviation of 0.4 on the calculated intensities (Vanneste et al., 2018). Earthquake depth and rupture orientation does not affect modelling outcomes as the BakunWentworth1997 IPE only takes into account epicentral distances. This is reasonable since faults in the LOFZ often have a surface trace and do not extend to great depths. Most earthquakes in the 2007 Aysén seismic swarm, for example, did not nucleate at depths larger than 10 km (Legrand et al., 2011).

In our analysis, we only considered crustal earthquakes with magnitudes ranging between 4.5 and 7.5, as lower magnitudes are unlikely to cause sufficient shaking for mass wasting to occur and higher magnitudes are uncommon in a crustal tectonic setting. Earthquakes related to the subduction megathrust were not taken into account here, as these are considered too distant (> 300 km from the Lago Pollux area) to cause significant shaking in the Lago Pollux area (Lazo, 2008). Moreover the absence of a coseismic deposit in the more trench-proximal Aysén Fjord advocates for an inland seismic source. Furthermore, we also do not consider intraslab earthquakes, as Lago Pollux and Lago Castor are situated merely 30-40 km north of the slab window that results from the subduction of the Chile Ridge (Russo et al., 2010), decreasing the likelihood of strong stress buildup in this area. Furthermore, extrapolation of the Slab2 model indicates that the slab depth at the location of lakes Castor and Pollux is around 140 km (Hayes et al., 2018), requiring minimum a ~M7 earthquake to cause shaking intensities of ≥VI½ (needed to trigger subaquatic mass-movements) based on an evaluation of historical Chilean events studied by (Astroza et al., 2005) and global intraslab earthquakes at a similar depth (USGS, 2023). Considering that a rupture to produce such M7 earthquake should be at least 50 km in length (Wells and Coppersmith, 1994), combined with the short distance to the slab window, an intraslab source is deemed unlikely in this area.

Instead of considering a grid of potential earthquakes, our modelling results are subsequently refined by considering known faults in the region (Figure 1a). Each of these faults was artificially cut into sections which can rupture individually as well as together following the guidelines in the Uniform California Earthquake Rupture Forecast, Version 3 (UCERF3; Field et al., 2014). We subsequently apply the magnitude scaling relationship by Wells and Coppersmith (1994) to each potential fault rupture, thus translating rupture length to earthquake magnitude, after which the Bakun and Wentworth (1997) IPE can again be applied as before.

# 4    Results

## 4.1    Bathymetry

220   Based on its bathymetry (Figure 2a), Lago Pollux can be divided into three subbasins: a deep northern subbasin, a deep central

subbasin and a shallow southern subbasin (NSB, CSB and SSB, respectively). The NSB and SSB are elongated, while the

CSB is wider and more circular. The deepest part of the lake, with a water depth of 56 m, is situated in the NSB. The transition

from the NSB to the CSB is marked by a clear shallower part (about 30 m water depth), incised by a small channel of about

40-45 m deep. The CSB is again deeper with a maximum depth of about 52 m. The SSB is a shallow subbasin, reaching depths

225   of maximum 25-30 m in its northern and central part and shallowing to ≤ 5-10 m in its southernmost part. This shallow

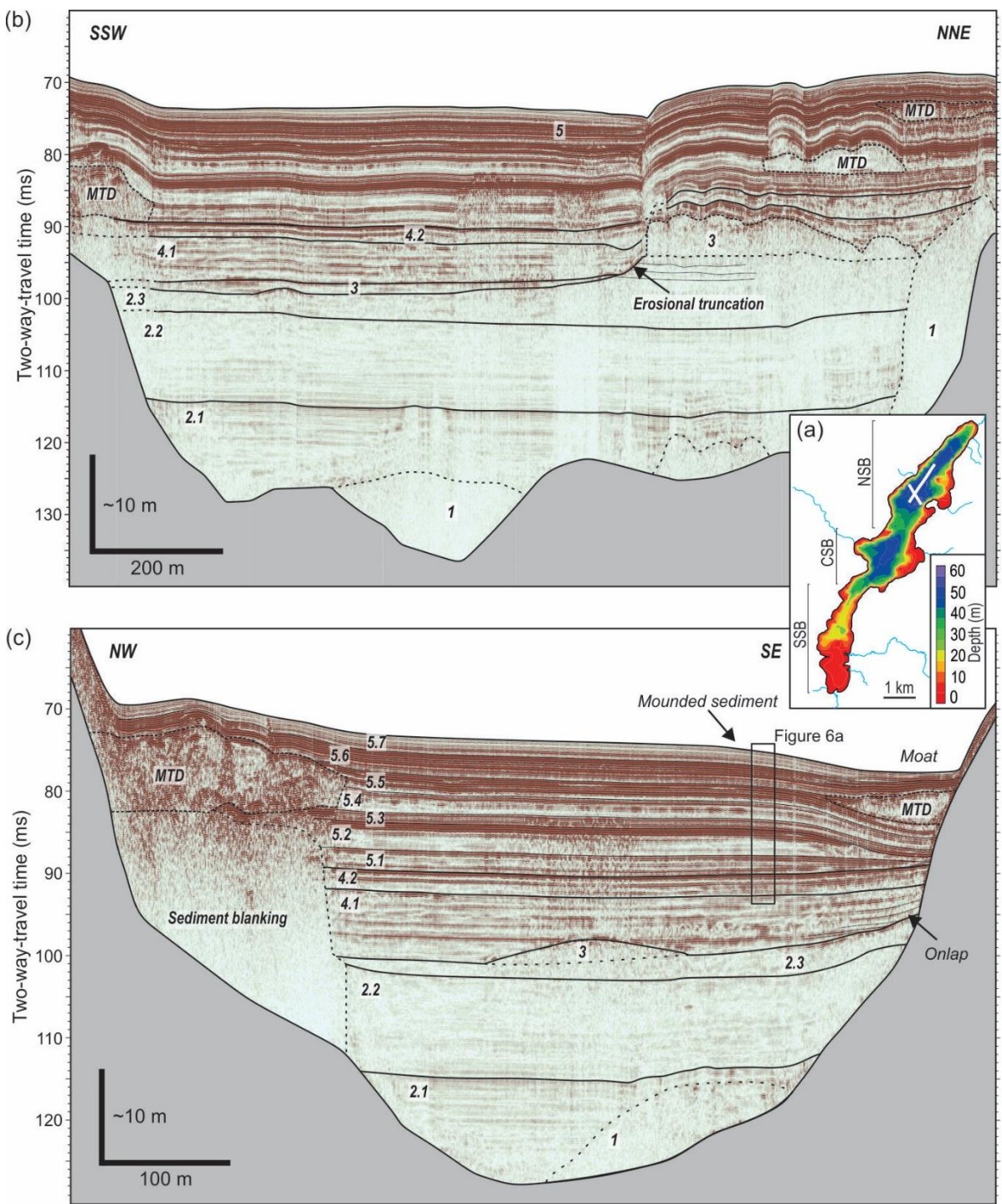

**Figure 2. (a) Bathymetry of Lago Pollux with indication of subbasins (NSB: northern subbasin, CSB: central subbasin, SSB: southern subbasin) and location of the two pinger seismic profiles (white lines). (b, c) Pinger seismic profiles of Lago Pollux with interpretations: acoustic basement (in grey), all seismic (sub-)units (numbers) and mass transport deposits (MTDs) are indicated (without interpretations: Figure S4).**

environment leads to low or even no penetration of the acoustic waves. The transition between the CSB and SSB is a shallow, narrow area of maximum 35-40 m deep. In a bay along the eastern shore of the NSB and CSB, depths of ≤ 5 to maximum 15 m are reached before evolving into the steeper slopes that flank the deeper basins. The slopes along the western shore of the NSB are steep. At the main outflow (western shore of the CSB) and the northern inflow (northern shore of the NSB), we observe a more gentle slope.

## 4.2 Geophysical data analysis of Lago Pollux

Five different seismic-stratigraphic units overlying the acoustic basement were identified on the seismic profiles of Lago Pollux (Figure 2b, c).

**Unit 1** is the lowermost seismic unit with a very variable thickness, ranging from ~24 m to less than 1 m. This unit only occurs locally and is mainly restricted to the deepest parts of the basin, filling local depressions. However, it is also present on bedrock highs throughout the NSB and CSB. It has a chaotic to transparent seismic facies.

**Unit 2** is the thickest sedimentary unit and is mainly observed in the NSB, where it has a maximum thickness of ~45 m in the central part, thinning towards ~36 m in the southern part. It is only sporadically present in the CSB with varying thickness of 20 to 40 m. This unit is characterized by continuous, (sub-)parallel and generally low-amplitude reflections. It shows a ponded geometry, filling up the deep subbasins, and does not drape the slopes. Three subunits can be identified based on subtle differences in reflection amplitude. The first subunit (Unit 2.1) shows generally higher-amplitude reflections compared to Unit 2.2 and 2.3. The seismic reflectors of Unit 2.1 are onlapping the acoustic basement or Unit 1, when present. At the top of Unit 2.3, erosional truncations can be observed on both the western and eastern slopes in the NSB and CSB (Figure 2b), making the top-boundary an unconformity.

**Unit 3** is a distinctive unit in which two different seismic facies can be identified: a chaotic to transparent facies and a hummocky facies. The chaotic-transparent facies is present in patches in the northern and central parts of the NSB (Figure 2b) and only sporadically observed in the CSB. Both the upper and lower boundaries form a clear unconformity with the surrounding seismic units. The hummocky facies, consisting of small higher-amplitude, discontinuous reflections, is only observed in the southern part of the NSB, which is the deepest part of the paleobasin (top of Unit 2). It has a maximum thickness of ~3 m and, in contrast to the chaotic facies, it shows a conformable upper and lower boundary.

**Unit 4** is typically thicker in the NSB compared to the CSB (up to ~23 m and ~16 m thick, respectively). However, unlike the underlying seismic units, it is observed throughout the entire basin. Its seismic facies is composed of subparallel to parallel continuous reflections. The unit is subdivided into two subunits, separated by a high-amplitude reflection, which is in turn covered by a chaotic to transparent facies with ponding geometry (Figure 2b). The whole unit shows an onlapping basal contact with Unit 2 and Unit 3 (when present). As a result, Unit 4.1 is more basin-focused than Unit 4.2, which is more widely distributed across the lake.

**Unit 5** is the uppermost seismic unit with a thickness of ~11 m in the deepest parts, showing continuously stratified seismic reflections that drape the previous unit. Towards the south-eastern slopes, the seismic reflections show a convergent facies, leading to a thinner sediment package (up to 6-8 m). Several thin subunits, seven in total, are identified throughout the basin based on differences in their reflection amplitude (Figure 2c). Additionally, multiple chaotic deposits can be identified in Subunits 5.1, 5.2 and 5.5. All chaotic deposits in Unit 5.1 and Unit 5.2 are located in the central and northern part of the NSB, except one which is found at the transition of the CSB to the SSB. Most deposits are located at the foot of the slope, with the exception of one deposit that is positioned towards the deeper areas of the central part of the NSB. The chaotic deposits in Unit 5.5 are mostly located along the western slope in the central and southern part of the NSB and the entire CSB. However, different deposits are identified along the eastern slope of the NSB and one at the eastern slope of the central CSB. The deposits along the eastern and western slopes show a very different seismic facies and morphology. Deposits along the western slope have a very chaotic facies and display successive frontal thrusts. Furthermore, seismic blanking of the underlying strata can

occur (Figure 2c). On the other hand, the deposits along the eastern slopes show a more transparent and less chaotic facies (Figure 2c). Additionally, they form relatively thin, lens-shaped sediment bodies located at the foot of the slope, thinning towards the central part of the subbasin. At the same horizon (top of Unit 5.5) in the deepest part of the basin, thin deposits with a transparent facies and ponding geometry can be identified (Figure 2b, c).

### 4.3    Lago Pollux and Lago Castor lithology

#### 4.3.1    Short cores

The short sediment cores (~60 cm), POL02 and CAST01 from Lago Pollux and Lago Castor, respectively, consist of laminated light to dark brown diatomaceous mud. We identify intercalations of multiple coarser and darker layers and interpret them as tephra layers. Overall, a low MS signal is recorded for the laminated mud, but four thicker tephra layers of 0.5 - 1.5 cm are marked by peaks in the MS values. Throughout the cores, and especially at the base of the cores, multiple smaller peaks are observed, indicating various smaller tephra beds. A correlation between both cores, and thus both lakes, is made based on the tephra layers and corresponding MS signal.

#### 4.3.2    Lago Castor U5.2 and 5.5 turbidite

The Lago Castor long sediment core consists mostly of grey and light to dark brown laminated mud and is described by Van Daele et al. (2016). Three turbidites were identified, of which two occur in the top 7.5 m of the core: in Unit 5.2 and 5.5. The turbidite in Unit 5.5 is located ~15 cm below a 20 cm thick tephra layer, interpreted by Van Daele et al. (2016) as the H2 Hudson Volcano tephra layer. Here, we revisit this turbidite and find that it has a very similar sedimentary composition as the background sediment and corresponds to what Van Daele et al. (2015) describe as type 1 lacustrine turbidites (LT1), resulting from sediment remobilization on hemipelagic slopes. Multiple MTDs were identified on that same horizon in the seismic profiles from Lago Castor (Van Daele et al., 2016).

### 4.4    Sedimentological analyses of Aysén Fjord

#### 4.4.1    Color and reflectance spectroscopy

The MD07-3117 core consists mostly of light to dark brown mud and is partly bioturbated (Wils et al., 2018). Section IX of the core (Figure 3) clearly shows a darker (lower brightness, L*) layer with a strong increase in radiodensity and $nRABA_{400-560}$ values around 11.02 m depth. Wils et al. (2020) identified this layer as a turbidite related to a megathrust earthquake. Additionally, a lighter-colored (increased brightness, L*) layer of about ~10 cm thick is present between ~11.15 and ~11.23 m, which is also marked by an increase in radiodensity, albeit less pronounced, but lower $nRABA_{400-560}$ values compared to the background sediment. Furthermore, the $R_{590}/R_{690}$ ratio shows higher values compared to those in the remainder of the studied core section, even higher than those observed in the turbidite. This layer was not recognized as an event deposit by Wils et al. (2020). Below this interval, at 11.26 and 11.31 m throughs in $nRABA_{400-560}$ values, corresponding to peaks in $R_{590}/R_{690}$ index and L*, can be explained by reworking of the sediments from the lighter-colored layer through bioturbation, and are thus post-depositional in nature. A correlation plot between the $nRABA_{400-560}$ and $R_{590}/R_{690}$ values shows three different

clusters: one corresponding to the background sediment samples, one to the turbidite sediments and one to the lighter-colored layer sediments (Figure 4a), indicating variations in source material.

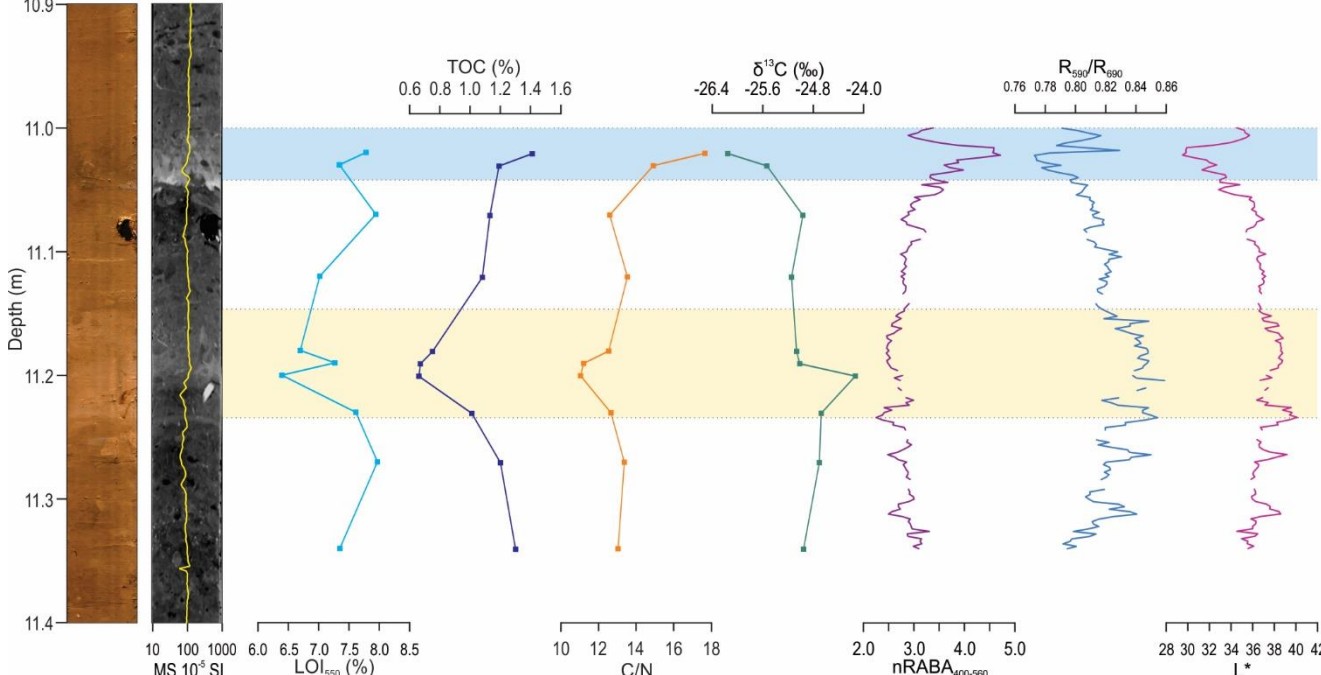

**Figure 3. Digital image and CT-scan of a portion (section IX, 10.9-11.4 m) of the MD07-3117 sediment core from inner Aysén Fjord, with bulk organic geochemistry and spectrophotometry data (from left to right): organic matter content (LOI550), total organic carbon content (TOC), carbon-nitrogen ratio (C/N), 13-carbon isotope fraction (δ 13C), normalized relative absorption band area between 400-560 nm (nRABA400-560), reflectance ratio between 590 and 690 nm (R590/R690) and brightness (L*). The turbidite (11.0-11.04 m) identified by Wils et al. (2020) and the lighter-colored layer (11.14-11.24 m) are indicated by blue and yellow bands, respectively. Gaps in the spectrophotometry data are due to previous sampling (holes) in the core.**

### 4.4.2    Bulk organic geochemistry

Throughout the studied core section, $LOI_{550}$ values are rather low and less than 8 %. The lowest values occur in the lighter-colored layer (6.4-7.6 %), while the turbidite shows higher values (7.3-7.8 %) compared to the background sediment. A similar

trend can also be seen in the TOC and C/N results (Figure 3). In the lighter-colored layer, TOC values only reach about 0.66 % and the C/N ratio is about 11. For the background sediment, the TOC content varies between 1.08-1.30 % and the C/N ratio between 13.0-13.5. The turbidite has a markedly higher TOC content of 1.41 % and a higher C/N ratio of 17.6. The opposite trend can be seen in $\delta^{13}C$ values, where values of -26.2 ‰ are reached at the base of the turbidite and -24.1 ‰ in the lighter-colored layer. However, these samples were at the limit of the measurement capability of the instrumentation, resulting in an

uncertain accuracy of the reported $\delta^{13}C$. Lastly, a positive correlation is present between the TOC content, $LOI_{550}$ and $nRABA_{400-560}$ values (Figure 4b). Generally it is assumed that the $LOI_{550}$ is two times the TOC-content, as typically ~50 % of organic matter is composed of organic carbon (Pribyl, 2010). Here, the slope of the linear regression between the $LOI_{550}$ and TOC values shows that ~72 wt% of the organic matter is organic carbon, while the intercept indicates that $LOI_{550}$ overestimates the organic matter content with ~6 %. This means other components are also present, such as water in clays and oxides,

explaining the apparent higher $LOI_{550}$ values. This is also reflected in the $nRABA_{400-560}$ and $R_{590}/R_{690}$ correlation plot (Figure 4a), where the $nRABA_{400-560}$ is a proxy for the TOC (Figure 4b; Vandekerkhove et al., 2020). As the $R_{590}/R_{690}$ values are a proxy for clastic material, it can be concluded that the lighter-colored layer contains more clastic components and less organic matter compared to the background sediment and turbidite (Figure 4a).

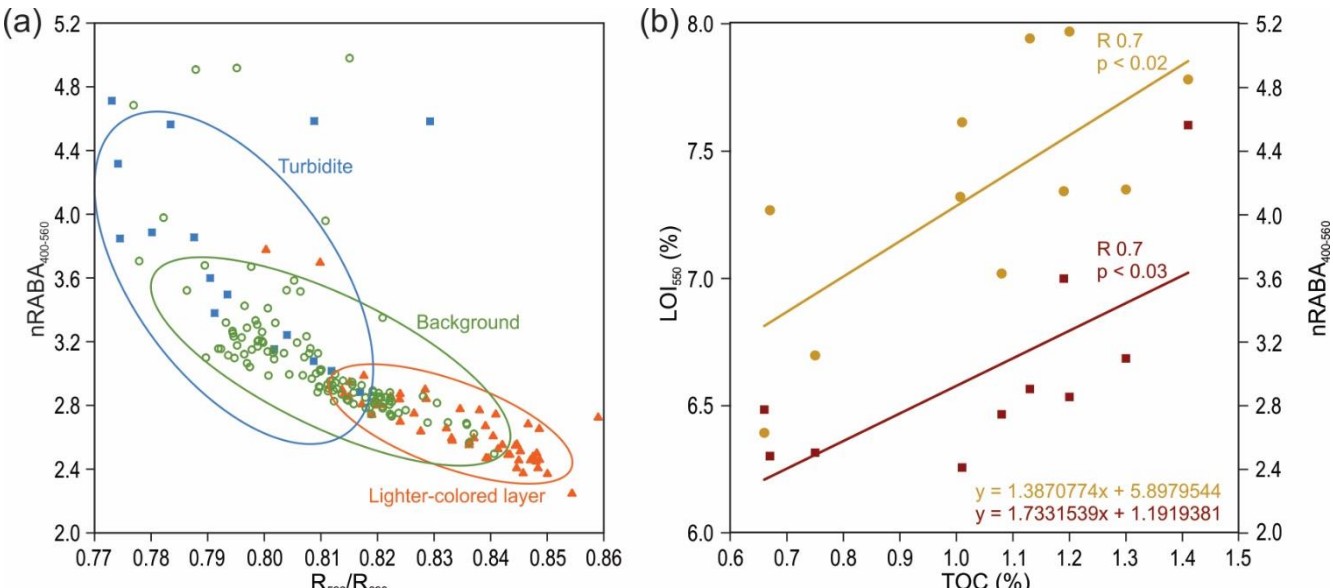

**Figure 4. (a) Correlation of the normalized relative absorption band area between 400-560 nm (nRABA400-560) and reflectance ratio between 590 and 690 nm (R590/R690) showing three distinct clusters: background sediment (green circles), turbidite (blue squares) and lightered-colored layer (orange triangles). (b) Correlation between total organic carbon (TOC) content with nRABA400-560 (red squares) and organic matter content (LOI550) (yellow circles).**

### 4.4.3   Grain size

According to the Udden-Wentworth classification, sediments in the studied core section can be described as silt, ranging from very fine silt in the lighter-colored layer to fine silt in the background sedimentation and medium silt at the base of the turbidite (Figure 5a). Throughout this core section, the sorting values are rather low and generally below ~3 μm. However, two peaks of poorer sorting up to ~5 μm can be seen at the base of the lighter-colored interval. In the turbidite, a slight increase in geometric sorting values is seen from the base towards the tail, indicating the coarse-grained base is slightly better sorted then

the finer sediments of the tail (Figure 5b). Three different EMs ($R^2 = 0.9912$) can be identified from the grain-size distributions of our 35 samples, two of which correspond well with the two finest EMs defined by Wils et al. (2020) for the complete MD07-3117 core (Figure 5c, d). The first population (EM0) consists of very fine material with a mode of 2.94 μm and is most abundant

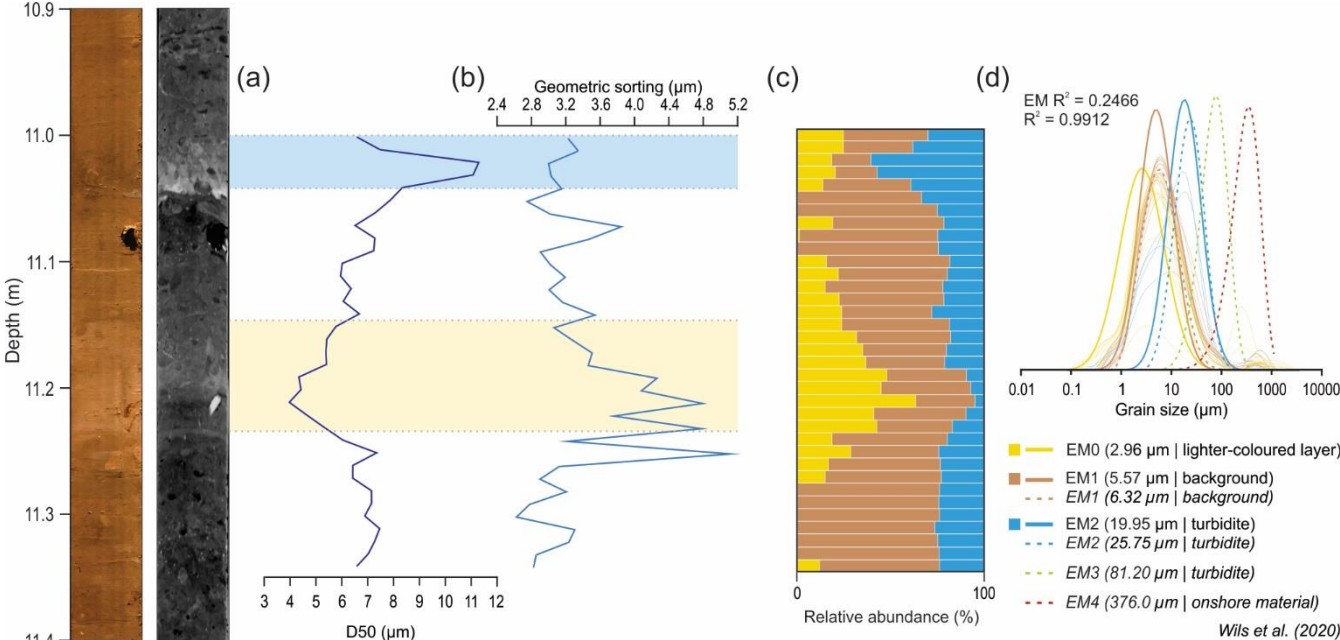

**Figure 5. Digital image and CT-scan of a portion (10.9-11.4 m) of the MD07-3117 sediment core from inner Aysén Fjord, with grain-size data including D50 values (a) and geometric sorting (b). (c) Relative abundance plot of the different end members identified in the portion of the MD07-3117 sediment core (EM0, EM1, EM2). (d) Plotted end member distributions of this study (full lines) with end member distributions from Wils et al. (2020) (dotted lines).**

in the lighter-colored interval (Figure 5c). The second (EM1) shows a larger grain-size mode of 5.57 μm and corresponds to the background EM described by Wils et al. (2020). Lastly, an end-member with a coarser mode of 19.95 μm is identified,

corresponding with the finest turbidite EM identified by Wils et al. (2020), and is thus also mostly present in the turbidite (Figure 5c).

## 5    Discussion

### 5.1    Sedimentary history of Lago Pollux

A similar seismic stratigraphy as Lago Castor is expected in Lago Pollux due to their proximity and similar geomorphological

position (i.e., both lakes are located on the same plateau). The units will be described and correlated to Lago Castor based on the interpretations of Van Daele et al. (2016).

Based on similarities with the lowermost units of sedimentary infills of other glacial lakes and fjords (e.g., Heirman et al., 2011, 2012; Ndiaye et al., 2014; Van Rensbergen et al., 1998), the chaotic deposits of Unit 1 in Lago Pollux can be interpreted as glacial till deposits (Figure 2b, c). Moreover, similar deposits were identified in Lago Castor, where they date back to MIS2

or MIS4 (Van Daele et al., 2016). This indicates that a glacial cover was present over Lago Castor and Lago Pollux during these glacial times.

During deposition of Unit 2, the sedimentary environment likely evolved to that of a sub- or proglacial lake characterized by more regular, glaciolacustrine sedimentation (e.g., Heirman et al., 2011; Ndiaye et al., 2014; Van Rensbergen et al., 1998; Figure 2b, c). This is also observed in Lago Castor, where a varved proglacial sedimentary unit was identified in the sediment

core (Van Daele et al., 2016). Unit 2 shows the same seismic characteristics and geometry in Lago Pollux and Lago Castor, allowing to interpret them as varved sediments as well. In Lago Castor Unit 2 is much thicker (up to 78 m) compared to Lago Pollux (up to 45 m), possibly indicating an earlier onset in sedimentation. Additionally, a thinning of the unit is observed from the NSB to the CSB in Lago Pollux, thus suggesting a deglaciation from NE to SW, in line with the general model for deglaciation of the Patagonian Ice Sheet (Davies et al., 2020).

The dispersed character of Unit 3, in combination with its transparent to chaotic facies, is distinctive for mass transport deposits (MTDs) (e.g Moernaut and De Batist, 2011). Based on the stratigraphy, this widespread mass wasting occurred around the same time as an erosional event, which is marked by the erosional features found at the top of Unit 2.3 (Figure 2b). Van Daele et al. (2016) also identified such an erosional event at the same seismic-stratigraphic level in Lago Castor, which was interpreted as erosion during lake level lowstand after a rapid drainage around > ~20 kyr BP. The model proposed by Van

Daele et al. (2016) predicts a drainage of Lago Castor through the narrow northern and central Lago Pollux subbasins. In Lago Pollux, the observed erosion may thus be related to the discharge event itself, where the exposed slopes subsequently or coevally failed, producing the observed MTDs.

After the event, the very low lake level rose gradually as shown by the ponding morphology and onlapping continuously-laminated seismic facies of Unit 4.1 (Figure 2c). Since deposition of Unit 4.2, sediments drape the entire lake floor and are no

longer limited to the deeper parts of the basins. A similar trend was also found in Lago Castor (Van Daele et al., 2016). Furthermore, in both lakes, a discontinuous high-amplitude reflection can be identified within this seismic-stratigraphic unit, in Lago Pollux covered by a chaotic/transparent facies with a ponding geometry (Figure 2b). In Lago Castor, this deposit is attributed to the presence of a ~60 cm thick volcanic ash layer that was identified in the sediment core between 9.17-8.58 m and interpreted as the Ho Hudson Volcano tephra (Van Daele et al., 2016). Considering the proximity of both lakes and the

dominant westerly wind directions (Gilli et al., 2005), the discontinuous high-amplitude reflections in both lakes are considered to be correlated and represent the base of the Ho tephra layer.

The most recent seismic-stratigraphic unit, Unit 5, in both Lago Pollux and Lago Castor shows draping and continuously laminated sediments, occasionally interrupted by MTDs (Figure 2b, c). The divergent seismic facies and the presence of moats

at the SE flanks in Lago Pollux (Figure 2c) are also present in Lago Castor, where they are interpreted as sediment drifts linked
to SWW activity since ~16.75 kyr BP (Van Daele et al., 2016). The seven subunits identified in Lago Castor are also seen in
Lago Pollux (Figure 2c, 6a). Van Daele et al. (2016) related the high-amplitude reflections separating the subunits to the
presence of tephra layers. Indeed, when comparing the short sediment cores from both lakes, representing the uppermost
sediments of Unit 5.7, a very similar sedimentation can be seen, with an almost identical pattern of MS peaks, which represent
the thin tephra layers (Figure 6b). As high-amplitude reflections correspond to tephra layers, it is not surprising that the subunits
in Unit 5 correspond between both lakes. Apart from Ho, the most prominent tephra layer identified in the sedimentary infill
of Lago Castor (top reflector of Unit 5.5) was interpreted as the H2 Hudson Volcano tephra deposit (Van Daele et al., 2016).
Hence, this tephra deposit is likely responsible for the strong reflector draping the MTDs at the top of Unit 5.5 in Lago Pollux
(Figure 2b, c, 6a). As expected, analysis of the seismic and sediment core data shows that Lago Pollux and Lago Castor
experienced a very similar sedimentary evolution.

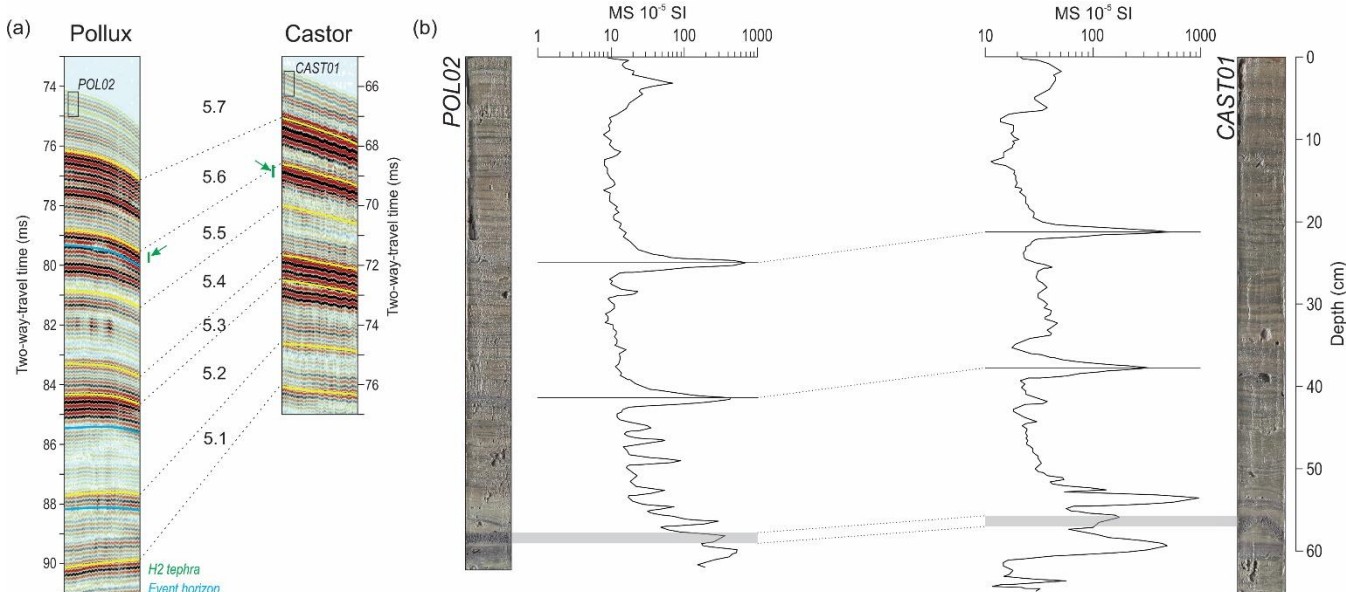

**Figure 6. (a) Seismic section of Unit 5 showing all seven subunits identified on a pinger seismic profile in Lago Pollux (Figure 2b) and Lago Castor (after Van Daele et al., 2016). (b) Correlation of two short cores from Lago Pollux (POL02) and Lago Castor (CAST01) with plotted MS data, showing two distinct peaks. At the base, a ~1.5 cm thick tephra in both cores is indicated in grey.**

**5.2     Synchronous mass wasting and a catchment response**

Towards the central part of the basin in Lago Pollux, the ponded, seismically transparent deposit associated with the MTDs of
Unit 5.5 is interpreted as a turbidite (e.g. Leithold et al., 2019; Praet et al., 2017; Schnellmann et al., 2005; Van Daele et al.,
2015). At the same seismic-stratigraphic horizon in Lago Castor, several MTDs were identified, correlating with a thin turbidite
in the sediment core (Van Daele et al., 2016). We interpret that the presence of these event deposits with multiple MTDs at
the same stratigraphic horizon (i.e. with 0.2 m accuracy) in these hydrologically open, neighboring lakes is not coincidental,
and hence, that they represent the same event. According to the synchronicity criterion (Adams, 1990; Schnellmann et al.,
2002), the observation of synchronous mass-wasting in two separate lakes points to a single, regional trigger mechanism for
this event, most likely earthquake shaking. The event occurred before the H2 Hudson Volcano eruption, as the MTDs are
covered by the associated tephra deposit. The age difference between the turbidite and H2 tephra in the sediment core from
Lago Castor is ~350 years (Van Daele et al., 2016) (Figure 8), which is thus assumed to be identical in Lago Pollux. The H2
tephra deposit was also found in Aysén Fjord, albeit with a slightly older age range (Wils et al., 2018, 2020) (Figure 8),
enabling a correlation between all three sedimentary records.

To verify whether the inferred earthquake responsible for mass wasting in Lago Castor and Lago Pollux also affected Aysén
Fjord, the results of grain-size analysis, organic geochemistry and spectrophotometry measurements on the section of the core

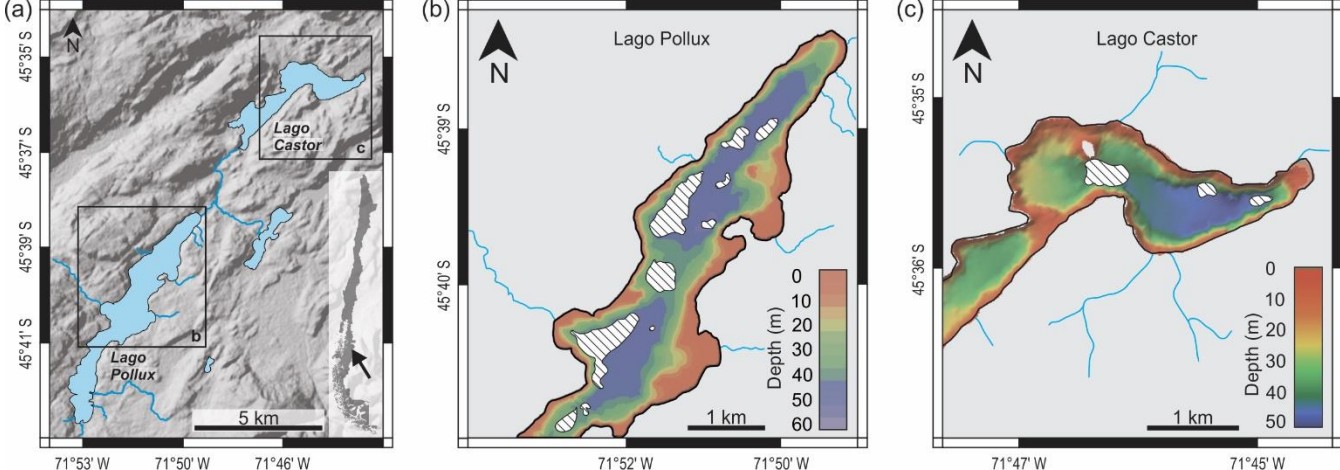

**Figure 7. (a)** SRTMGL1N terrain data (NASA JPL, 2013) of the area around Lago Pollux and Lago Castor. **(b, c)** Spatial distribution of the mass transport deposits (MTDs; the hatched shapes) identified in Unit 5.5 under the H2 tephra deposit for both Lago Pollux **(b)** and Lago Castor **(c)**. Note that the distribution of MTDs in the southern part of Lago Pollux could not be mapped due to shallow bathymetry and limited seismic penetration, and that no MTDs were identified in the southwestern part of Lago Castor.

below the H2 deposit were analyzed. Three deposits appear distinctively different from the background sediment: two turbidites related to megathrust earthquakes by Wils et al. (2020), and the lighter-colored interval (Figure 3). While the absolute age of neither of these deposits matches with that of the turbidite in Lago Castor, the relative age difference between the base of lighter-colored interval and the H2 deposit (~385 years) is very similar to the age offset observed for the turbidite and H2 tephra in Lago Castor (~359 years; Figure 8). This confirms that the observed mass-wasting in Lago Castor and Lago Pollux

was not related to a megathrust earthquake (i.e., no MTDs or turbidite in Aysén Fjord), but rather to a local shaking event located towards the east, closer to Lago Castor and Lago Pollux than to Aysén Fjord. Nevertheless, this event seems to have resulted in lighter-colored sediment input in the fjord (Figure 8). The overall low organic matter content in this layer ($LOI_{550}$ values; Figure 3), combined with the higher $R_{590}/R_{690}$ values (Figure 3) indicate a higher clastic sediment input (Figure 4a). The additional very fine silts to clays (EM0) in this layer (Figure 5) further suggest that this increased clastic component is a

result of additional input of clastic, mainly in the form of fine-grained material. We interpret this anomalous additional clastic sediment input as a catchment response signal: significant mass wasting in the Aysén Fjord catchment, consisting of volcanic rocky soils and low amounts of vegetation, could lead to a temporary increase in clastic sedimentation input. The long-distance fluvial transport from the catchment area towards the final deposition in Aysén Fjord explains the finer grain-size trend observed in this layer (Figure 5). The low C/N values in this catchment response (Figure 3) are somewhat surprising, as one

would expect increased C/N values resulting from the added terrestrial organic matter. However, such low C/N ratios for terrestrial organic matter can be explained by decomposition processes of organic matter in the catchment soils. These processes comprise microbial immobilization of N-rich material accompanied by remineralization of C (Meyers and Ishiwatari, 1993; Shanahan et al., 2013). Furthermore, the andosol soils in the Aysén Region are already low in organic matter due to weathering processes on the hilly slopes (Ellies, 2000; Gut, 2008), which explains the modest effect. The local high $\delta^{13}C$ values

in the lighter-colored layer (where also the grain-size effect is strongest; Figure 3, 5), can also be explained by the decomposition processes of soil organic matter. In the $CO_2$ resulting from microbial respiration of soil organic matter, the lighter isotope of carbon ($^{12}C$) is preferred to be incorporated, leaving the heavier $^{13}C$ isotope behind (Shanahan et al., 2013). In other words, the lighter-colored layer seems to contain more decomposed organic matter combined with a larger clastic component compared to the background sediments. In this case, the increase in clastic sedimentation input lasted for a

maximum of ~50 years (based on the age model; Figure 8), but likely less, as the catchment response would have increased the overall sedimentation rate, but tis is hard to quantify with the available data. Such timeframe seems plausible because at present, 17 years after the 2007 Aysén $M_w$ 6.2 earthquake, which caused multiple landslides in the Aysén Fjord area (Sepúlveda et al., 2010), not all of the then-exposed slopes have been re-vegetated; and increased erosion rates thus likely persist. If the

earthquake epicenter was located more to the east (which is expected based on the observed combination of shaking imprints, and will later on be confirmed by ground-motion modelling) where the climate is drier, this process would be even slower.

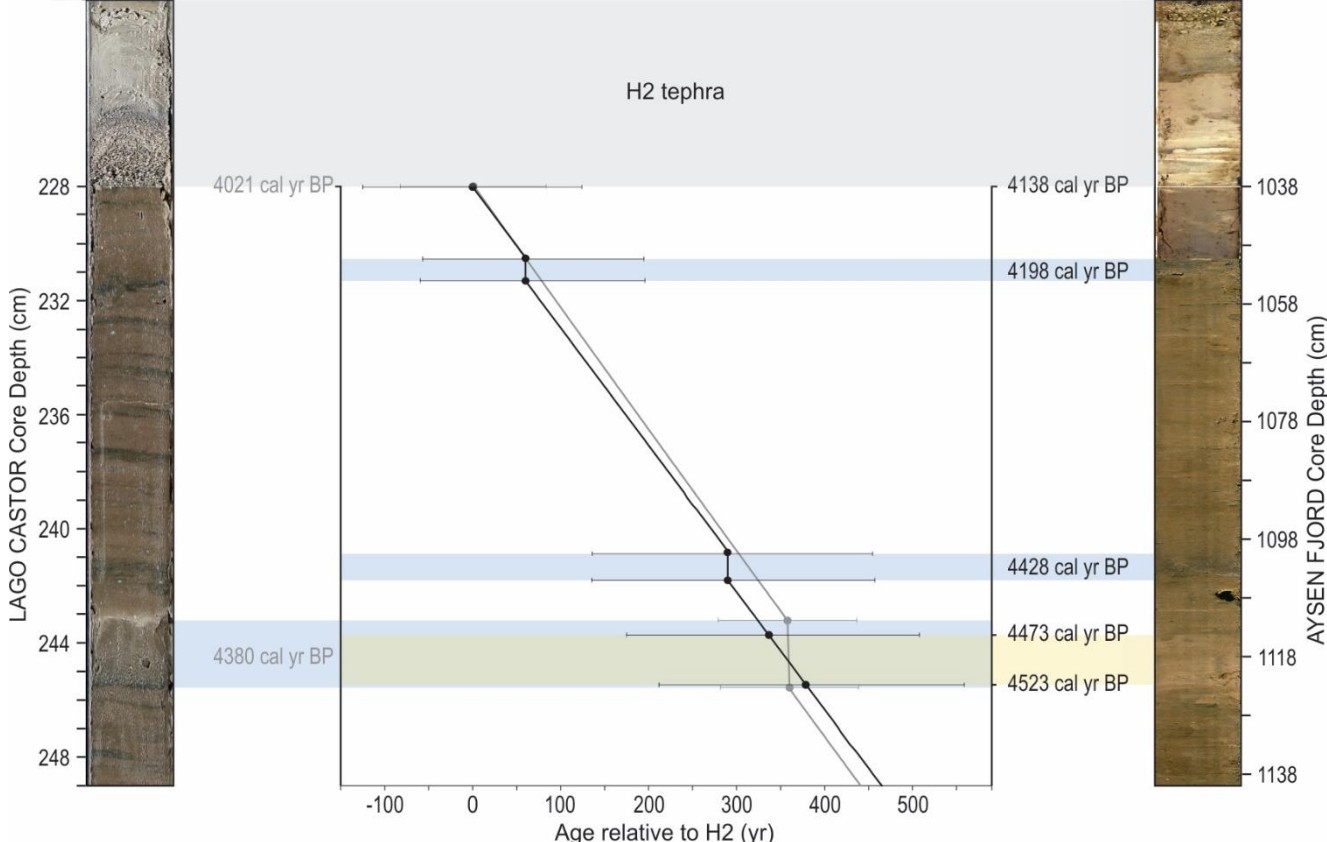

**Figure 8. Correlation figure between the sediment core retrieved from Lago Castor (left) and Aysén Fjord (right) (core pictures not to original scale). The graph shows an age-depth model relative to the H2 Hudson Volcano tephra deposit (4.09-3.61 cal yr BP; Naranjo and Stern 1998; grey band). For each indicated age on the graph, the 95% confidence interval is plotted. Turbidites in both cores are indicated by blue bands and the lighter-colored interval by a yellow band. The correlation shows an overlap between the deposition of the turbidite from Lago Castor and lighter-colored layer from Aysén Fjord.**

### 5.3 Locating the earthquake source

For each of the study sites, the minimum and/or maximum intensity of seismic shaking is determined based on the sedimentary evidence (Table 1). In Lago Castor, all identified MTDs attributed to the earthquake ~4400 years ago are located along the lake slopes (Figure 7). These consist of hemipelagic sediments, and no post-seismic catchment response (pointing to onshore mass-wasting, Howarth et al., 2012) was identified (Figure 8). This implies that the shaking intensity at the time of the earthquake must thus have exceeded VI½ (i.e., threshold for failure of lateral lacustrine slopes covered with hemipelagic sediments), but remained well below VIII (i.e., threshold for onshore landslides in the catchment). In Lago Pollux, similar, albeit more widespread, MTDs are observed (Figure 7), but it is unclear whether these consist of hemipelagic sediment or rather originated onshore due to the lack of a sediment core. A minimum shaking intensity of VI½ must thus have been achieved, but a maximum threshold cannot confidently be assigned. Local shaking intensities in Aysén Fjord are considered not to have exceeded VI, as not even a failure of the Río Aysén delta could be identified, which would have resulted in a turbidite similar to those that have been related to megathrust earthquakes (Wils et al., 2020).

**Table 1: Overview of all earthquake evidence used as input for ground-motion modelling in all study sites**

|  | **Positive evidence** | **Negative evidence** |
|---|---|---|
| **Lago Castor** | ≥ VI½ | < VIII |
| **Lago Pollux** | ≥ VI½ | / |
| **Aysén Fjord** | / | < VI |

| Río Aysén catchment | > VII½ in predefined area | / |
| --- | --- | --- |

However, the up to ~50 year period of enhanced sediment input (Figure 8) does indicate that a significant amount of landslides must have occurred in the Rio Aysén catchment. This provides an additional constraint for the ground-motion modelling, as intensities of VII½ must have been exceeded at least in a part of the Río Aysén catchment. The maximum size of the area affected by landslides during an earthquake increases with magnitude (Keefer, 1984; Rodríguez et al., 1999). This relationship also holds true for observations in the Aysén Fjord area following the 2007 $M_W$ 6.2 earthquake (Sepúlveda et al., 2010), and can thus be used to estimate the area that would have been affected by landslides during the ~4400 cal yrs BP event. In general, no landslides can be observed for earthquakes with an $M_W$ lower than 4, while an earthquake with $M_W$ 8 results in a maximum affected area of about 100,000 km² (Keefer, 1984; Rodríguez et al., 1999). In practice, this implies that the entire Río Aysén catchment, with a size of about 13,000 km², could have been affected. Considering the ~50 year duration of increased sediment input in Aysén Fjord as a result of this earthquake, it is unlikely that onshore landslides occurred in an area smaller than 100 km². However, it is impossible to determine exactly what part of the catchment experienced high enough shaking to produce onshore landslides as this depends on a number of factors such as earthquake depth and geological site conditions, influencing the presence of slide-susceptible sites and seismic attenuation. Therefore, we test a range of areas varying between 100 and 10,000 km² (within the Río Aysén catchment) and use the minimum intensity threshold of VII½ to limit the range of possible sources that are considered in the probabilistic method: earthquake ruptures that cannot produce this intensity of seismic shaking (based on forward application of the IPE) in the predefined minimum portion of the Río Aysén catchment are discarded (Figure 9).

As expected, modelling results show that, for any of the predefined area values, the probabilities for a crustal fault to cause the considered combination of positive and negative evidence are zero for any earthquake in the western part of the catchment,

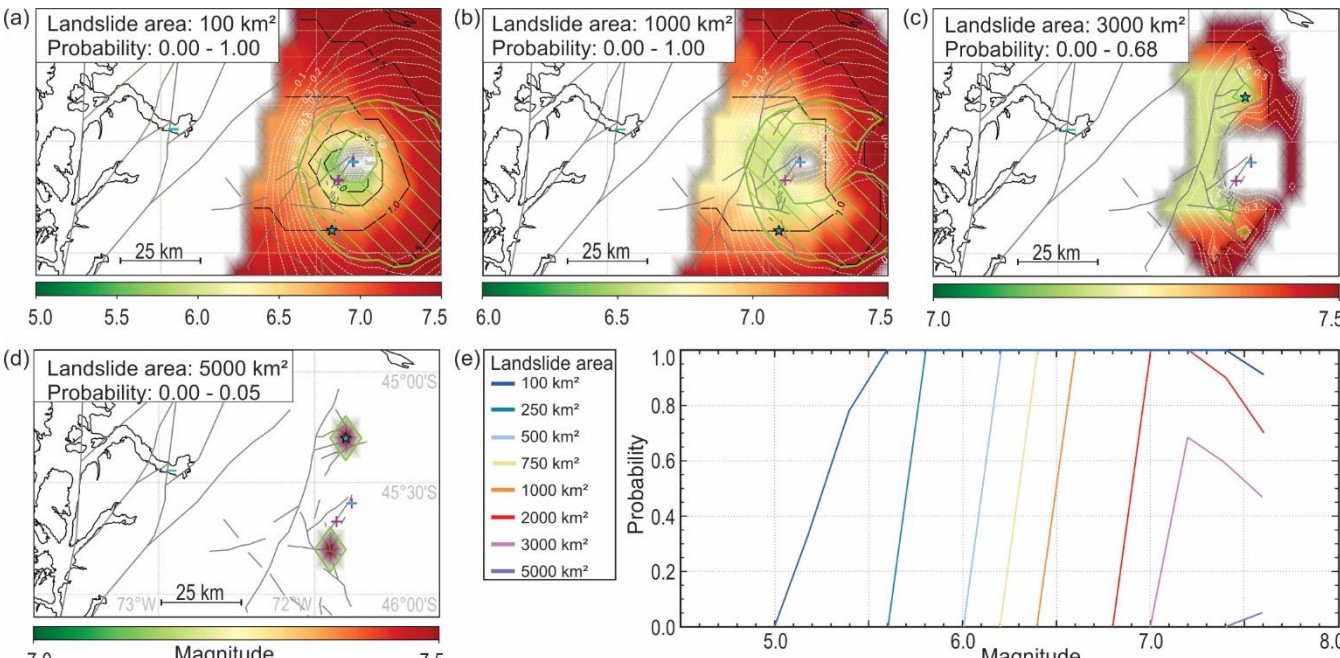

**Figure 9. (a-d) Maps showing the area (plotted in Figure 1a) in which an earthquake could have caused the observed pattern of sedimentological imprints In Lago Pollux, Lago Castor and Aysén Fjord (positive and negative evidence), considering an area of 100 (a), 1000 (b), 3000 (c) or 5000 (d) km² that was affected by landslides. Fault traces are mapped after SERNAGEOMIN (2003). The magnitude with the highest probability at each location is given in a green-to-red color scale. White dashed lines correspond to the probability; green line and enclosed hatched area mark the region where the probability is less than 0.1 below the maximum probability, highlighting the most likely epicentral region (indicated by a star). Sites of positive (plus) and negative (minus) evidence are shown in semitransparent magenta and cyan colors, respectively. Note that sites with both positive and negative evidence show a bluish color. (e) The maximum probability of an earthquake in the study area (independent of its location) with a magnitude ranging between 5 and 7.6 to have caused the observed pattern of sedimentological imprints In Lago Pollux, Lago Castor and Aysén Fjord (positive and negative evidence), considering a range of minimum areas (100-5000 km²) in the Rio Aysén catchment that were affected by landslides.**

close to Aysén Fjord (Figure 9a, b), meaning that the earthquake must have occurred close to Lago Castor and Lago Pollux.

This immediately rules out the potential for an LOFZ-rupture, or rupture along the Azul Tigre or Río Mañihuales Fault, to have caused the observed mass-wasting, and indicates that the earthquake must have been hosted by one of the faults in the eastern part of the Río Aysén catchment. Moreover, it is highly unlikely that this earthquake caused landslides over an area of 3000 km² or larger in the Río Aysén catchment, as probabilities then rapidly decrease to zero (Figure 9c). For an area of 100 km², maximum probabilities can be observed for earthquakes with an $M_W$ between 5.5 and 7.5 (Figure 9a, e). Probabilities

remain high for areas of up to 2000 km², although the minimum earthquake magnitude then increases to 7.0 (Figure 9e). The considered landslide area thus mostly constrains the minimum magnitude required to result in the observed mass-wasting. When taking into account the approximate fault traces in the area, reasonable probabilities (> 50%) are only obtained for areas around 1000 km² or smaller and for earthquakes with magnitudes below 7, regardless of the considered landslide area (Figure 10). This indicates that most of the faults are in an unfavorable location or not sufficiently long to produce the right range of

shaking intensities at the study sites and simultaneously affect a large portion of the Río Aysén catchment. The minimum magnitude is again constrained by the size of the landslide area within the Río Aysén catchment, with an $M_W$ of 5.6 for 100 km², increasing to 6.4 for 500 km² and 6.8 for 1000 km² (Figure 10h). For higher magnitudes, probabilities decrease significantly and the most likely responsible fault also shifts westward, away from Lago Castor and Lago Pollux and towards Coyhaique.

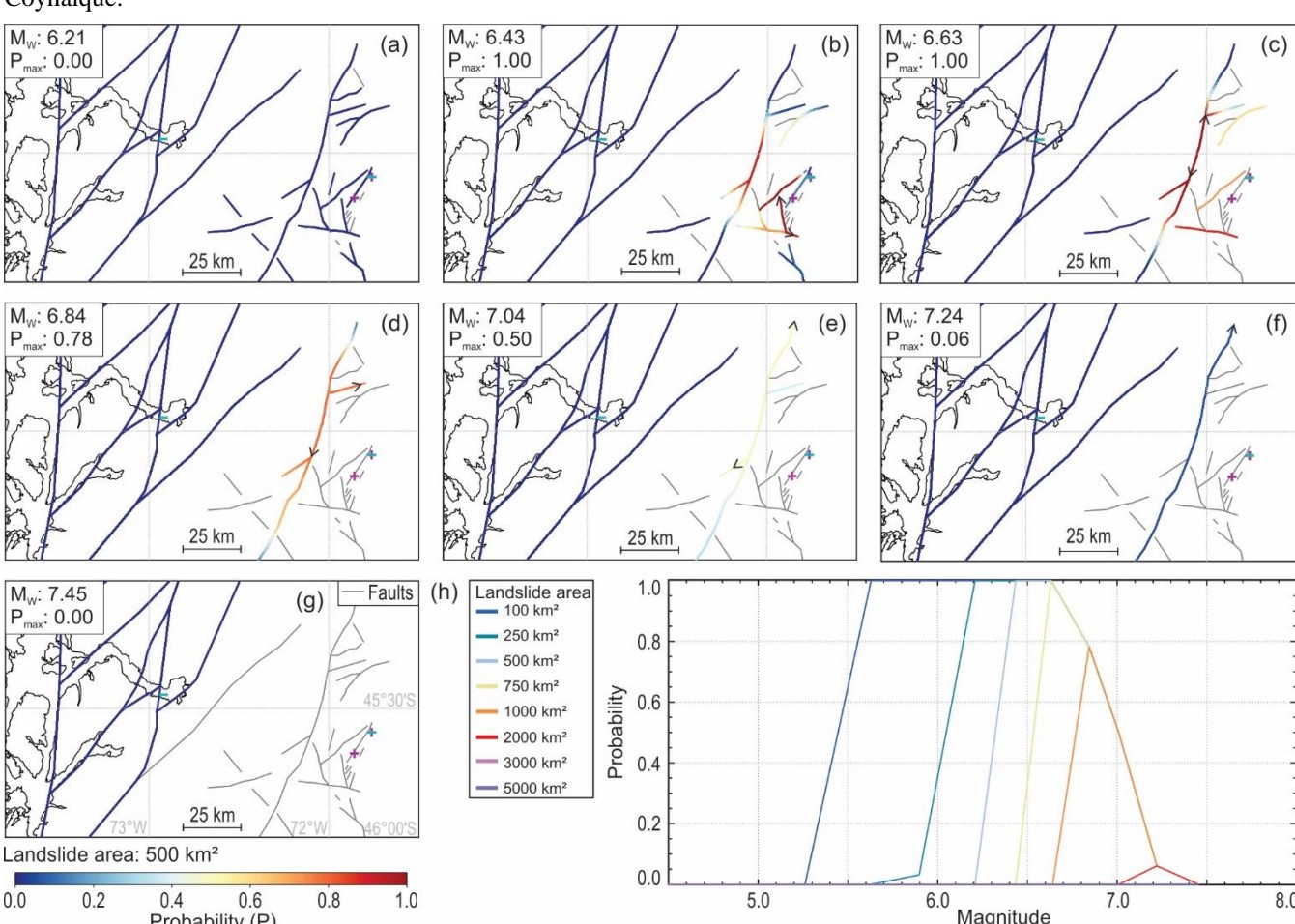

**Figure 10. (a-g)** Maps showing all faults in the study area (plotted in Figure 1a), color-coded according to the probability that an earthquake of magnitude 6.21 (a), 6.43 (b), 6.63 (c), 6.84 (d), 7.04 (e), 7.24 (f), 7.45 (g) on each of the fault sections is capable of causing the observed spatial distribution of sedimentary earthquake evidence and by considering an area of 500 km² that has been affected by landslides. Fault traces are mapped after SERNAGEOMIN (2003). Arrows indicate the limits of the rupture trace resulting in the highest probability. Grey fault traces are not sufficiently long to cause a rupture with the tested magnitude. Sites of positive (plus) and negative (minus) evidence are shown in semitransparent magenta and cyan colors, respectively. Note that sites with both positive and negative evidence show a bluish color. **(h)** The maximum probability of an earthquake on any of the faults in the study area with a magnitude ranging between 5 and 7.6 to have caused the observed pattern of sedimentological imprints In Lago Pollux, Lago Castor and Aysén Fjord (positive and negative evidence), considering a range of minimum areas (100-5000 km²) in the Rio Aysén catchment that were affected by landslides.

We thus reach the highest (maximum) probabilities (1.0) for scenarios with a magnitude around 6.5. For the slightly shorter ruptures (M6.43; Figure 10b), the N-S trending Cordon La Galera Fault (southwest of Lago Pollux; Figure 1c) and the NE-SW trending Pollux Fault (the northwest of Lago Pollux and Castor; Figure 1c) are the most likely candidates. For somewhat larger ruptures (M6.63; Figure 10c), the NNE-SSW trending unnamed fault that runs through the Río Simson valley in Coyhaique would be the main candidate. While we cannot further assess which of these faults ruptured during the 4.4 kyr
earthquake, a mapping effort to assess if any of these faults offset Holocene sediments (De La Cruz et al., 2003) may shed more light on this.

### 5.4 Earthquake hazard in the Coyhaique region

Even though a future, similar relatively high-magnitude crustal event poses a seismic hazard to the region, these events do not seem to occur frequently. The sedimentary infill of Lago Castor and Lago Pollux do not reveal any other similar, large-scale
mass-wasting events throughout the Holocene, and there are no other traces of similar catchment responses in the sediment core from Aysén Fjord. Only a few MTDs in the middle of Unit 5.2 in Lago Pollux that are likely synchronous with a turbidite in Lago Castor (Van Daele et al., 2016) point to a similar event around 13 kyrs BP. However, as this is beyond the maximum age of the Aysén record, the sedimentary evidence is spatially insufficiently distributed to obtain meaningful modelling results. Hence, the local faults near Coyhaique do not seem to be very active, or do not produce sufficient shaking to be recorded in
lake sediments. Indeed, several of the faults are too short to produce M>6.5 earthquakes which would cause strong shaking over a large area (Figure 10d). Furthermore, on the Lago Castor seismic profiles that cross the Castor Fault, we did not observe fault offset within Unit 5. Nevertheless, one of the most likely rupture scenarios for the 4.4 kyr earthquake is a rupture on an unnamed fault that runs directly through the city center of Coyhaique (Figure 10b-d), the capital and most populated city of the region. Even though such earthquakes seem to have a very low average recurrence rate since the deglaciation, the low-
probability but high-intensity hazard (up to intensity IX, Figure 11) may be relevant especially for critical infrastructure, such as hospitals and fire-departments which are crucial for a quick response, especially in such a remote region.

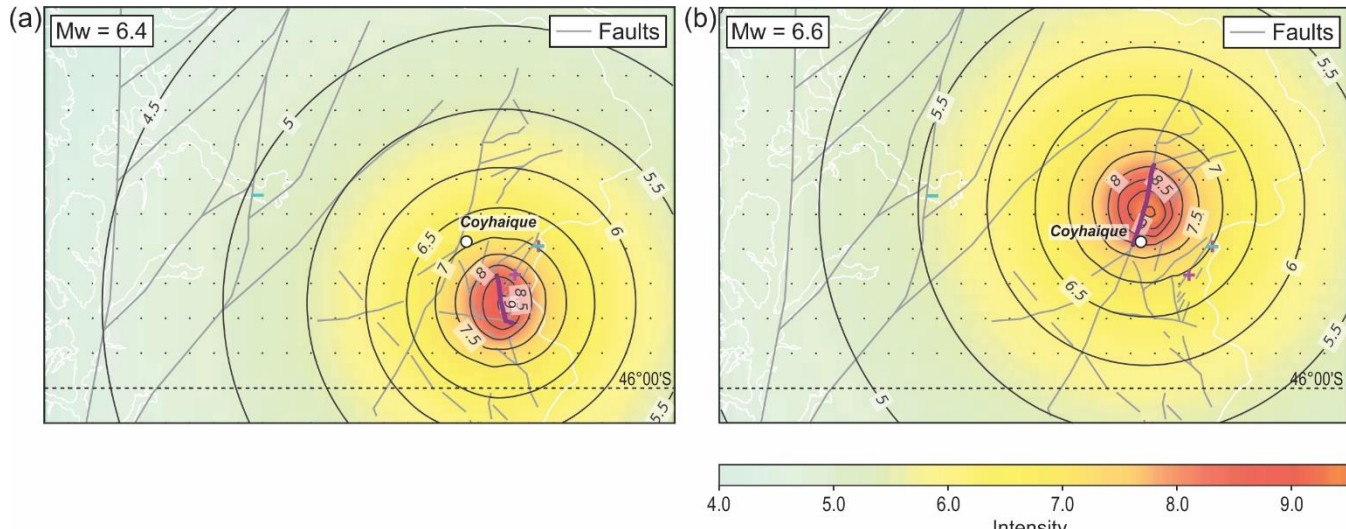

**Figure 11. Maps (plotted in Figure 1a) showing macroseismic intensities (MMI scale) predicted by the BakunWentworth1997 IPE for a magnitude 6.4 earthquake along the Cordon La Galera Fault (a) and 6.6 earthquake along the unnamed fault through the city center of Coyhaique (b), both of which are considered as likely scenarios to cause the observed pattern of lacustrine paleoseismic evidence around ~4400 cal yrs BP. Fault traces are mapped after SERNAGEOMIN (2003). Sites of positive (plus) and negative (minus) evidence are shown in semitransparent magenta and cyan colors, respectively. MMI = Modified Mercalli Intensity, IPE = intensity prediction equation.**

The low earthquake probability in the Coyhaique region contrasts to that near the LOFZ faults, for which 5 events, including the 2007 earthquake, have been identified throughout the Holocene (Wils et al., 2018). All of these events produced major mass-wasting in Aysén Fjord, and are the result of moderate-magnitude earthquakes on either of the LOFZ fault branches in
the vicinity of the fjord (Vanneste et al., 2018). An additional event recently identified in Laguna Esponja, located ~45 km

north of the city of Coyhaique and on the Río Mañihuales Fault (Figure 1b), around 166 CE (63 BC – 345 AD; Fagel et al., 2023) is potentially also related to activity of the LOFZ, the Azul Tigre or Río Mañihuales Fault. Around this time, a coarse-grained turbidite has also been identified in the sedimentary infill of Aysén Fjord (youngest EM3 in Wils et al., 2020; 170 – 350 AD; note that our hypothesized correlation is different to that proposed by Fagel et al., 2023), potentially associated to onshore mass-wasting as well – albeit less pronounced compared to the aforementioned events (Wils et al., 2020). If these deposits indeed correlate, this would indicate the presence of a 6[th] event along one of the fault branches of the LOFZ, merely a few tens of years following one of the predecessors of the 2007 $M_W$ 6.2 event (Wils et al., 2018), or alternatively a rupture of either the Azul Tigre or the Río Mañihuales fault.

## 6    Conclusions

Analysis of seismic reflection profiles of Lago Pollux reveals its sedimentary history and allows for a correlation with neighboring Lago Castor. Based on the published age depth models of Lago Castor and Aysén Fjord, a correlation between these three records was established. This correlation shows that synchronous mass-wasting occurred around ~4400 cal yrs BP in both lakes, while Aysén Fjord was not directly affected by subaqueous landslides but did experience the sedimentary effects of a ~50 years long catchment response as a result of major onshore mass-wasting in its catchment. Evidence for this catchment response was obtained by multiproxy analysis of a section in a sediment core retrieved from inner Aysén Fjord. Grain-size results showed that a portion, seen as a lighter-colored layer, showed a distinct finer-grained component. Furthermore, a lower organic matter content was found in this deposit, matching the already organic-poor andosol soils in the catchment area. Ground-motion modelling provides constraints on the location and magnitude of the triggering earthquake. For an area affected by landslides between 100 and 1000 km², an earthquake rupture along a local fault, with a magnitude range of 6.3-7.0 and intense shaking, is the most-likely scenario. A megathrust-related source mechanism or a rupture along the LOFZ as source can be ruled out. It can thus be concluded that the seismic hazard in the Aysén Region is not only restricted to the main, large-scale tectonic structures, but could also come from smaller local faults in the area, posing a low-probability, but high-intensity hazard that could be relevant for the hazard of the city of Coyhaique, with most critical infrastructure in the region. Further paleoseismic studies in small Patagonian lakes or on the faults themselves could aid in a better understanding of the seismic hazard of these local faults.

**Author contribution**

Reflection-seismic interpretations of Lago Pollux and sedimentological analysis of the Aysén Fjord sediment core were carried out by Morgan Vervoort, aided by Maarten Van Daele and Katleen Wils. The Aysén Fjord sediment core was retrieved during a cruise for which Catherine Kissel was chief scientist. Marc De Batist led the research projects that provided funding for the fieldwork and analysis. Mario Pino and Roberto Urrutia provided logistical support enabling the lake fieldwork. Analysis of the short cores in Lago Pollux and Lago Castor was carried out by Clara Paesbrugge, aided by Maarten Van Daele. Earthquake modelling was performed by Katleen Wils, applying a code developed by Kris Vanneste and Katleen Wils. Morgan Vervoort prepared the manuscript, with contributions from all co-authors.

**Acknowledgements**

This research was funded by the Research Foundation Flanders (FWO-Vlaanderen CHILT project) (FWO G.0778.09). K. Wils is funded by the Research Foundation – Flanders (FWO), grant 12ZC422N. We thank K. De Rycker, A. Peña and O. Wuendrich for invaluable, technical help on the field. We thank C. Paesbrugge for the sedimentological analysis of the Lago Castor and Lago Pollux short cores. We further thank the Isotope Bioscience Laboratory (ISOFYS) of Ghent University for

the organic geochemical analysis. S&P Global Software Geoscience Package (Kingdom version 2019) is acknowledged for
their educational user license.

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
