# Peer review of "Co- and postseismic subaquatic evidence for prehistoric fault activity near Coyhaique, Aysén Region, Chile"

_EGUsphere, 2024_

## Referee Comment (RC1)

A. **GENERAL COMMENT**

I enjoyed reading the manuscript written by Vervoort et al, which presents a co- and post-seismic record at Lago Pollux, and shows that along with previously published studies (Lago Castor; Van Daele; 2016 and Aysén Fjord, Wils et al. 2020), local faults can also produce significant earthquakes. The authors use a 'common' methodology in lake palaeoseismology but brings here ground motion modelling to derive the most likely magnitude and source fault, which is quite innovative and interesting.

The manuscript is well written and organised, although I found it sometimes relies heavily on the findings of Van Daele et al. 2016 and Wils et al. 2020, which can overshadow its original contributions and make it difficult to follow without prior knowledge of these papers. I think making some more clarification or adding supplementary information in the methods and results sections could help reducing this issue.

Another point that concerned me a bit was the lack of important key information about how the age models were made. Although not from their study, the correlation between the Lago Castor and Aysén Fjord sediment cores is an important point in their discussion section. Their argument is based on synchronicity and then on the age models, which can be very different from one method to another. I then suggest please to write explicitly how they were modelled in the respective articles, at least briefly, which modelling settings and inputs.

Regarding the last part of the manuscript, I think that the results and discussion regarding the source fault model are very interesting, but the readability and visual quality of figures 9 and 10 do not allow readers to take full advantage of them. I believe that improving these figures will greatly enhance the impact of your study.

Therefore, I recommend minor/moderate revisions to address these concerns before considering final publication.

B. **SPECIFIC COMMENTS**

**Introduction**

Line 30: I think you should also mention that we can also observed co-seismic deformation with sediment cores on each part of faults. e.g.:

- Beck, C., Campos, C., Eriş, K. K., Çağatay, N., Mercier de Lepinay, B., & Jouanne, F. (2015). Estimation of successive coseismic vertical offsets using coeval sedimentary events–application to the southwestern limit of the Sea of Marmara's Central Basin (North Anatolian Fault). Natural Hazards and Earth System Sciences, 15(2), 247-259.
- Gastineau, R., De Sigoyer, J., Sabatier, P., Fabbri, S. C., Anselmetti, F. S., Develle, A. L., ... & Gebhardt, A. C. (2021). Active subaquatic fault segments in Lake Iznik along the middle strand of the North Anatolian Fault, NW Turkey. Tectonics, 40(1), e2020TC006404.

**Study area**

Line 57: Sometimes you use "Figure. 1", sometimes "Fig.", please check the style of the journal and change accordingly.

Figure 1: If you know it, please say which type of fault you have (normal, reverse, strike-slip). Fonts are hard to read.

**Method**

Line 101: Please remove "(very)": it does not help the reader and is not quantitative.

Line 114: It could be nice to add the CTD profile in the supplementary information.

Line 122: "The present study focusses on sections VIII and IX of the core (9.5-12 m depth)". This is not a useful information because in the article of Wils et al., 2020; I could not find these sections' names. Better to refer to the depths maybe? Or to add the scheme of the composite core in supplementary information (SI) to help the reader to check the information you provide.

Line 159: Delta can also collapse spontaneously, maybe you could explicit that.

Line 163: I find this +1/2 unit uncertainty a bit arbitrary but this is true I don't have better to propose. Why not 1 then? Can you in one sentence argue about that?

Line 179: Please add a reference for all the fault characteristic values you provide.

**Results**

Line 188: "Figure 2c", the figures should appear in the chronological order. Same line 242 where we jump from figure 2 to figure 6b.

Line 201: To help the reader, you could add a scale in meters on the right of your figure 2. Besides, you mention an acoustic velocity of 1450 m/s in the water to produce the bathymetric map. What about the sediment? Is the velocity assumed constant? Is it based on the MSCL measurements for the topmost part of the seismic where you have a sediment core? Which value do you use? We missed such important information in the method.

Figure 2:
- It is not super easy to make our own interpretation of the seismic profile due to the quality of the PDF. Can you please add uninterpreted profile in the SI at least, please? For example, the unit 3 I would have been happy to see a zoom of it.
- In Figure 2c, add a scale please.
- You talk in the method of both a centipede sparker and geopulse spinger source system data, but only present two profiles from the spinger source system, why?
- Please provide in the map the location of all the seismic profile that you have, and why did you choose to present these ones? If you don't present the sparker data please remove from the method section.

Lines 247-255: This section is mainly based on the results of other studies. You should either create a section where you summarise the results of other studies, or reformulate your results so that they are better highlighted here.

Line 257: "Section IX"; again, you refer to a numbered section, but I don't get this information elsewhere so it doesn't help to understand your text.

Line 266: I would remove "clearly"

Line 271: Provide the values for the lowest and highest values in brackets.

Figure 5c, there is no colour scale to read the plot.

**DISCUSSION**

Line 313: "In Lago Castor Unit 2 is much thicker (up to 78 m) compared to Lago Pollux," please give again here the thickness in Lago Pollux.

Line 325: I think it is important to show explicitly those onlaps on the figure.

Figure 7: the caption should say (b) Lago Pollux and (c) Lago Castor and you must then write a caption for (a). Please add scales. If you have all those seismic profiles why not displaying the thickness of those MTDs? The maps in b and c are quite empty: you could also add some info on the slopes to help to better constrain the uncertainties that you have "To accommodate for variability in preconditioning factors, such as slope angle or availability of sediment". Also, maybe add the rivers etc to see where we have more sediment accumulation? I think this figure can be significantly improve, you have a lot of useful information that would help in terms of interpretations and reasoning.

Figure 8: I have a concern regarding the approach and this figure. We do not have in the method section any information regarding the age modelling part. I know both cores are from another study; but as it is the main part of your argumentation you should be clear on how the age model were performed in both papers. If they are not run using the parameter then I would be happy to see how it is with rBacon for example. Please be more explicit about that in your manuscript.

Line 389: "are located along the lake slopes". More or less, at least it is hard to see that on the actual figure.

Line 391: "cfr"?

Line 392: I thought it was VI1/2 + 1/2 = VII? from your method line 158. Then I am a bit lost.

Figure 9: In general, the fonts of this figure are too small. The maps are hard to read. The isolines are hardly perceptible. etc. This need to be significantly changed please. Same for the GPS coordinates, they are too small. I can unfortunately can't review it properly.

Line 423: Figure 9c I presume? But I can't see what you mean.

Line 437: I see your point, and I am not working in subduction zone, but 30 km is not a lot for such strong earthquakes to my opinion?

Figure 10: I have several questions regarding this model.

- From where do you take the fault map?
- Is there any study on the faults themselves (e.g., trenches)? Data regarding the movement of the faults?
- You will not have the same wave propagation with a strike slip or a normal fault. How sure are you about this tectonic map? Here you mentioned for example "Grey fault traces are not sufficiently long to cause a rupture with the tested magnitude"; Then you should maybe mention it. I feel there are some key information missing regarding tectonics. And I have the same remark as Figure 9, this figure it generally too small and really hard to read.

C. **TECHNICAL CORRECTIONS**

Line 38: Please remove '~'.

Line 104: Missing dot after "receiver"

Line 106: typo "m in" and not "m)in"

Caption Figure 1: area and not are.

Line 252: add a space after "(2015)".

---

## Author Comment (AC1)

**Date:** 29/03/2024

**Manuscript number:** EGUSPHERE-2024-8

**Title of article:** Co- and postseismic subaquatic evidence for prehistoric fault activity near Coyhaique, Aysén Region, Chile

**Name of Corresponding Author:** Morgan Vervoort

**Email Adress of the Corresponding Author:** Morgan.Vervoort@UGent.be

**RC1:** Renaldo Gastineau

**RC1 review:** Indicated in blue

A. GENERAL COMMENT

I enjoyed reading the manuscript written by Vervoort et al, which presents a co- and post-seismic record at Lago Pollux, and shows that along with previously published studies (Lago Castor; Van Daele; 2016 and Aysén Fjord, Wils et al. 2020), local faults can also produce significant earthquakes. The authors use a 'common' methodology in lake palaeoseismology but brings here ground motion modelling to derive the most likely magnitude and source fault, which is quite innovative and interesting.

The manuscript is well written and organised, although I found it sometimes relies heavily on the findings of Van Daele et al. 2016 and Wils et al. 2020, which can overshadow its original contributions and make it difficult to follow without prior knowledge of these papers. I think making some more clarification or adding supplementary information in the methods and results sections could help reducing this issue.

Another point that concerned me a bit was the lack of important key information about how the age models were made. Although not from their study, the correlation between the Lago Castor and Aysén Fjord sediment cores is an important point in their discussion section. Their argument is based on synchronicity and then on the age models, which can be very different from one method to another. I then suggest please to write explicitly how they were modelled in the respective articles, at least briefly, which modelling settings and inputs.

Regarding the last part of the manuscript, I think that the results and discussion regarding the source fault model are very interesting, but the readability and visual quality of figures 9 and 10 do not allow readers to take full advantage of them. I believe that improving these figures will greatly enhance the impact of your study.

Therefore, I recommend minor/moderate revisions to address these concerns before considering final publication.

Thank you for your review. All comments are acknowledged and will be reviewed and discussed below. It is indeed true that the manuscript relies on previous research, therefore some lines will be added for clarification.

Furthermore, a few lines will be added on the age models. This is, the number of radiocarbon ages used and the type of material on the age models for both Lago Castor and Aysén Fjord. However, it is important to keep in mind that we made a correlation between both the Lago Castor and Aysén Fjord age model relative to the H2 tephra deposit, thus relying on sedimentation rates

rather than absolute ages. This implies that those details about the radiocarbon ages, material type and calibration should not be the main focus. However, the age models of Lago Castor and Aysén fjord will be added in the SI to illustrate that the sedimentation rates are very stable.

Lastly, both Figure 9 and 10 will be improved: 1) The map of Chile will be deleted and a box will be added on Figure 1 referring to the locations in Figure 9 and 10. 2) We will edit the figure to make room to enlarge the maps, improving the readability and visual quality of the images.

B. SPECIFIC COMMENTS

Introduction

Line 30: I think you should also mention that we can also observed co-seismic deformation with sediment cores on each part of faults. e.g.:

• Beck, C., Campos, C., Eriş, K. K., Çağatay, N., Mercier de Lepinay, B., & Jouanne, F. (2015). Estimation of successive coseismic vertical offsets using coeval sedimentary events–application to the southwestern limit of the Sea of Marmara's Central Basin (North Anatolian Fault). Natural Hazards and Earth System Sciences, 15(2), 247-259.

• Gastineau, R., De Sigoyer, J., Sabatier, P., Fabbri, S. C., Anselmetti, F. S., Develle, A. L., ... & Gebhardt, A. C. (2021). Active subaquatic fault segments in Lake Iznik along the middle strand of the North Anatolian Fault, NW Turkey. Tectonics, 40(1), e2020TC006404.

These references will be added to the manuscript.

Study area

Line 57: Sometimes you use "Figure. 1", sometimes "Fig.", please check the style of the journal and change accordingly. This will be updated so that it is consistent throughout the article (i.e. "Figure 1").

Figure 1: If you know it, please say which type of fault you have (normal, reverse, strike-slip). Fonts are hard to read. The type of fault is written in the text for the LOFZ (line 75) but will be added in the manuscript text for the local crustal faults (line 79). After the addition of an indicative rectangle for Figure 9 and 10 (see A. General Comment) and rivers in the Aysén catchment (see further), the figure and its fonts will be updated.

Method

Line 101: Please remove "(very)": it does not help the reader and is not quantitative. This will be removed.

Line 114: It could be nice to add the CTD profile in the supplementary information. Since this is not the focus of the article, only the reference to the original study will be added (i.e. Van Daele et al., 2016).

Line 122: "The present study focusses on sections VIII and IX of the core (9.5-12 m depth)". This is not a useful information because in the article of Wils et al., 2020; I could not find these sections' names. Better to refer to the depths maybe? Or to add the scheme of the composite core in supplementary information (SI) to help the reader to check the information you provide. A scheme of the composite core and core sections will be added in the SI, alongside the SI figure of the

Aysén Fjord age model (see A. General comment), and a reference to the data report from the original survey.

Line 159: Delta can also collapse spontaneously, maybe you could explicit that. Since the focus is on external triggering mechanisms, this will be added explicitly in the manuscript.

Line 163: I find this +1/2 unit uncertainty a bit arbitrary but this is true I don't have better to propose. Why not 1 then? Can you in one sentence argue about that? The ½ intensity unit is used because smaller increments constrain too much, and larger increments would imply that negative evidence has no effect on the outcome. The ½ increment is thus a good compromise, as was found by Kremer et al. (2017) and is also further explained in Vanneste et al. (2018). This will be added in the manuscript, including the explicit references.

Line 179: Please add a reference for all the fault characteristic values you provide. We hereby refer to the geological map of the area, an inline citation will be added.

Results

Line 188: "Figure 2c", the figures should appear in the chronological order. Same line 242 where we jump from figure 2 to figure 6b. Figure 2 will be set in chronological order: the bathymetry map is Figure 2a and the profiles Figure 2b, c. The reference to Figure 6 will be left out of this results section.

Line 201: To help the reader, you could add a scale in meters on the right of your figure 2. Besides, you mention an acoustic velocity of 1450 m/s in the water to produce the bathymetric map. What about the sediment? Is the velocity assumed constant? Is it based on the MSCL measurements for the topmost part of the seismic where you have a sediment core? Which value do you use? We missed such important information in the method. In Figure 2, on both seismic profiles a scale is visible, showing both a vertical estimation of 10 m of depth (and a horizontal scale). In line 112-113 it is noted that the acoustic velocities from Lago Castor were used to estimate the sediment thickness (with reference to the study on Lago Castor; Van Daele et al., 2016).

Figure 2:

- It is not super easy to make our own interpretation of the seismic profile due to the quality of the PDF. Can you please add uninterpreted profile in the SI at least, please? For example, the unit 3 I would have been happy to see a zoom of it. In the SI, for both profiles, the uninterpreted version will be added as a figure.

- In Figure 2c, add a scale please. A scale will be added on the bathymetry map in Figure 2.

- You talk in the method of both a centipede sparker and geopulse spinger source system data, but only present two profiles from the spinger source system, why? As mentioned in the methods, the sparker source resulted in a vertical resolution of ~0.5 m. As this resolution is low, the profiles were not used for the identification of the MTDs. However, these profiles were used to reconstruct the bathymetry and aid in the distribution (thickness/depth) of the different seismic units. This will now be clearly stated in the methods section.

- Please provide in the map the location of all the seismic profile that you have, and why did you choose to present these ones? If you don't present the sparker data please remove from the method section. The seismic profiles were not plotted on the bathymetry map in Figure 2, to not overshadow the bathymetry (as the whole lake was surveyed in a dense grid). However, a detailed

map of all the pinger and sparker profiles will be added in the SI. The two profiles shown in the manuscript are the best profiles to display the various seismic characteristics of the whole lake; i.e. the typical seismic facies of the units and the different MTD types and their presence in the different units.

Lines 247-255: This section is mainly based on the results of other studies. You should either create a section where you summarise the results of other studies, or reformulate your results so that they are better highlighted here. We will reformulate this in the manuscript to clarify that these results came from another study.

Line 257: "Section IX"; again, you refer to a numbered section, but I don't get this information elsewhere so it doesn't help to understand your text. A scheme of the composite core and sections will be added in the SI (see above).

Line 266: I would remove "clearly" This will be removed from the manuscript.

Line 271: Provide the values for the lowest and highest values in brackets. These values will be added.

Figure 5c, there is no colour scale to read the plot. The color scale is the same as shown in Figure 5d. However, this shall be made clear in the legend of the figure.

DISCUSSION

Line 313: "In Lago Castor Unit 2 is much thicker (up to 78 m) compared to Lago Pollux," please give again here the thickness in Lago Pollux. The value will be added to this line.

Line 325: I think it is important to show explicitly those onlaps on the figure. This will be indicated in Figure 2.

Figure 7: the caption should say (b) Lago Pollux and (c) Lago Castor and you must then write a caption for (a). Please add scales. If you have all those seismic profiles why not displaying the thickness of those MTDs? The maps in b and c are quite empty: you could also add some info on the slopes to help to better constrain the uncertainties that you have "To accommodate for variability in preconditioning factors, such as slope angle or availability of sediment". Also, maybe add the rivers etc to see where we have more sediment accumulation? I think this figure can be significantly improve, you have a lot of useful information that would help in terms of interpretations and reasoning. The caption will be updated. Thickness maps of the MTDs could not be made because of two main reasons: The sparker profiles have a vertical resolution that is too low (~0.5 m) to properly identify the MTDs. As the pinger and sparker profiles alternate (not every survey line is both a pinger and sparker profile), and based on the pinger profiles alone, the gridding of the reflectors would give too much of an error. On Figure 7b, c the rivers will be added, as well as the bathymetry, to aid in the interpretation and reasoning.

Figure 8: I have a concern regarding the approach and this figure. We do not have in the method section any information regarding the age modelling part. I know both cores are from another study; but as it is the main part of your argumentation you should be clear on how the age model were performed in both papers. If they are not run using the parameter then I would be happy to see how it is with rBacon for example. Please be more explicit about that in your manuscript. This information will be added to the manuscript (see A. General Comment).

Line 389: "are located along the lake slopes". More or less, at least it is hard to see that on the actual figure. Figure 7 will be updated (see comment Figure 7), to aid the reader while reading this section.

Line 391: "cfr"? This will be removed from the manuscript.

Line 392: I thought it was VI1/2 + 1/2 = VII? from your method line 158. Then I am a bit lost. By explaining this in the methods section (see above), we hope that this will be clear.

Figure 9: In general, the fonts of this figure are too small. The maps are hard to read. The isolines are hardly perceptible. etc. This need to be significantly changed please. Same for the GPS coordinates, they are too small. I can unfortunately can't review it properly. Figure 9 (and 10) will be modified to enhance the readability (see A. General Comment).

Line 423: Figure 9c I presume? But I can't see what you mean. Throughout this section, the references to Figure 9 will be updated and clearly point to a certain part of the figure.

Line 437: I see your point, and I am not working in subduction zone, but 30 km is not a lot for such strong earthquakes to my opinion? The slab window resulting from the subduction zone is 30-40 km away. However, the trench itself is located more than 300 km from the Lago Pollux area, making this too distant to cause significant shaking in the area. Moreover, the absence of a coseismic deposit in the more trench-proximal Aysén Fjord advocates for an inland seismic source. This explanation will be added to the manuscript to clarify.

Figure 10: I have several questions regarding this model.

- From where do you take the fault map? The fault map is a simplified version of the geological map of the area (SERNAGEOMIN, 2003). This reference will be added in the manuscript (and in the caption of Figure 9 and 10).

- Is there any study on the faults themselves (e.g., trenches)? Data regarding the movement of the faults? No; a simplified geological model was used.

- You will not have the same wave propagation with a strike slip or a normal fault. How sure are you about this tectonic map? Here you mentioned for example "Grey fault traces are not sufficiently long to cause a rupture with the tested magnitude"; Then you should maybe mention it. I feel there are some key information missing regarding tectonics. And I have the same remark as Figure 9, this figure it generally too small and really hard to read. Although the overall stress regime in the region is related to strike-slip faulting due to the oblique subduction of the Nazca plate below the South American plate, it is indeed true that individual faults in the region may exhibit some normal component of slip. This would indeed affect wave propagation and thus intensity distributions. However, the considered IPE relies on epicentral distances, and the modelling outcomes would thus not change when considering a different focal mechanism. To avoid any confusion, we will clarify this in the methods, and add some background on how earthquake magnitude scales with fault length (using a magnitude scaling relationship) – thus explaining why some faults are not capable of causing certain tested earthquake scenarios.

C. TECHNICAL CORRECTIONS

Line 38: Please remove '~'.

Line 104: Missing dot after "receiver"

Line 106: typo "m in" and not "m)in"

Caption Figure 1: area and not are.

Line 252: add a space after "(2015)".

All the above will be corrected in the manuscript.

---

## Author Comment (AC2)

**Date:** 09/04/2024

**Manuscript number:** EGUSPHERE-2024-8

**Title of article:** Co- and postseismic subaquatic evidence for prehistoric fault activity near Coyhaique, Aysén Region, Chile

**Name of Corresponding Author:** Morgan Vervoort

**Email Adress of the Corresponding Author:** Morgan.Vervoort@UGent.be

**RC2:** Pierre Sabatier

**RC2 review:** Indicated in blue

The paper of Morgan Vervoort et al. about « Co- and postseismic subaquatic evidence for prehistoric fault activity near Coyhaique, Aysén Region, Chile» investigate lake and fjord sediment records as a potential earthquake archive thanks to seismic stratigraphy and sediment cores with sedimentology and chronology data. The paper is well written and structured and result provided a new methodology to estimate earthquake location. However, I have to mains concerns 1/ some proxies used in Fjord Aysen (reflectance data) and some proxy interpretation (C/N) and 2/ the estimation of intensity threshold, how it was estimated and if it fluctuates through time with large implication on the medullisation part. Thus, I suggest major revision before to take into consideration for publication

Major comments:

1/Reflectance data: the author provide some proxy of TOC and mineralogical content derived from spectrophotometric data. The first one integrates the spectrum area with oxide contents and thus does not fully correspond to what was mentioned by the authors. This proxy could be an indicator of TOC but for that it needs to be calibrated by punctual analyses, and from Figure 3, it is obvious that this proxy does not fit with the TOC measurements. The use of this proxy is surprising, as many other proxies exist and are more robust for reconstructing TOC or at least chlorophyll content from this type of data. For the other proxy R590/R690 as a proxy of mineralogical content, yes it is use for like that by (Trachsel et al., 2010). I know that the USGS uses this proxy for mineralogical content but not for soft sediment with a high amount of organic matter, which is known to have a signal in this specific spectral range. As chlorophyll interacts with the spectrum at 670 nm, it is difficult to avoid integrating chlorophyll content into this proxy. L* is probably a better proxy for that, and a good comparison with R590/R690 is an argument. For well-established proxies, such as Chlorophyll, punctual analyses are not needed, but for others it is important, as they could be site dependent; thus, I strongly recommend that, as Trachsel said, "prior to interpreting the reflectance spectra, the general mineralogical composition and geochemistry of the sediment should be measured by established analytical methods (e.g., XRD)". I can also recommend to the authors to have a look on a recent review publication on hyperspectral data (containing visible data) : (Jacq et al., 2022). The comparison between spectral proxy and TOC or LOI try to be done in Figure 4 and we can said that it is not good with a high dispersion of the data and if the author provide the correlation coefficient and the associated p value it will be for sure not validate, this is why I recommend to try other better define organic matter proxy. You understand that I have some doubt about the use of these proxies and Figure 4A confirm this doubt because if R590/R690 is a proxy of mineralogical content why the turbidite and light coloured layer has not similar values.

Since we measured the OM (LOI$_{550}$) and TOC content, the spectrophotometric data have been used as supplementary proxy to further illustrate and highlight our results. In this study, we do not focus on the mineralogy. But, by using these proxies we aimed to illustrate that the lighter-colored layer has a higher mineralogic content and thus suggesting a more clastic input.

The R and P values will be shown on Figure 4b, and since the R = 0.7 and P < 0.03 (LOI in function of TOC), together with the low number data points, these results are considered significant.

Furthermore, we did not expect that the lighter-colored layer and turbidite had similar values. As is confirmed by our results, illustrating a different origin between the turbidite in Aysén Fjord and the lighter-colored layer.

2/The interpretation of C/N ratio is very strange for me. The decomposition of organic matter is likely present (probably no so strong knowing this could environement), but it actually changes the C/N ratio, may be, but for a part of this organic matter will not expect this change. It is very strange that the C/N ratio decreased with increasing terrestrial inputs in the fjord record, especially when compared with that of turbidites, for which the C/N ratio increased. Did you consider potential GLOF deposits in this fjord record because a GLOF deposit will present lower TOC content (Piret et al., 2021), greater fine terrestrial input and potentially some organic matter previously deposited in aquatic environments, thus with low C/N ratio… Is it possible to have GLOF in the catchment, such as in other Patagonian Fjords (Vandekerkhove et al., 2021)

We do agree that the C/N values were rather anomalous. However, by the fact that it does decrease in this layer, we interpreted it as more decomposed organic matter being present. Together with the spectrophotometric results, indicating a more mineralogic/clastic input, this shows that the layer is different from the background sediment, and the turbidite, for which the higher LOI, TOC and C/N (and lower δ$^{13}$C) values indicate more fresh organic matter.

We did not consider GLOF deposits, as this region has been deglaciated prior to the event: deglaciation started > 20 kyr BP (Van Daele et al., 2016) and the event is slightly older than the H2 tephra deposit (4.09-3.61 cal yr BP).

3/As few data about chromoly is presented in this paper (present in already published ones) it is important to specify if these lake system experience variations of sedimentation rate over time which could modify the sensitivity of lake to record earthquake event. If variation in the sedimentation rate occurred in the past, this could modify the availability of the sediment on the slope and thus the threshold to record earthquake with specific intensities will change also (Wilhelm et al., 2016; Rapuc et al., 2018).

The sedimentation rate has been very stable in both Lago Castor and Aysén Fjord throughout the Holocene. To illustrate this, the age models of both Lago Castor and Aysén Fjord will be added in the SI.

4/I have a main concern about the estimation of intensity threshold to record event deposit in this Fjord and lakes. If I well understand this estimation came from a comparison with a New Zeland sites and other worldwide? This threshold must be estimated from historical record and not record earthquakes on these sites, as there are already some papers published on these sediment sequences in which this intensity limit can be estimated to record or not earthquake… This threshold depends on many local parameters (faults, type of earthquakes, and lake parameters

such as sedimentation rate); thus, this threshold cannot be compared with what is already published worldwide. Without this precise estimation you cannot rule out the part about ground motion modelling. Maybe I do not understand something when I read the paper because I know that this team works well and made such estimation. Thus, if it is already estimated please add more clearly on the revised version. In addition, of course, this sensitivity could change over time in regard, for instance, to changes in the sedimentation rate, but additional information is needed; see the previous main comment.

Indeed, ideally, shaking threshold values are determined based on local calibration using negative evidence of historical earthquakes. However, especially for Lago Castor, no historical events are available to conduct such site-specific calibration. And therefore, average values based on the evaluation of global thresholds are used. This will be added in the manuscript, including in-line references to Vanneste et al. (2018) and Van Daele et al. (2020).

Minor comments:

All the minor comments were reviewed and changes were made in the manuscript.

L62: precise the type of Cretaceous rock

L66: Rio Simpson is not located on Figure 1

L179: how does you estimate the dip?

L210: for me on this figure it is just visible on the Eastern part

L214: CSB not presented, at least add it in supplementary

L220: I do not see the upper limit in Fig 2

L228: not in 5.4?

L242: Figure have to be presented in the right order, not 6 before previous ones.

L243: interpretation have to move after

Figure 5: may be add a contour plot it could be useful to identify grain size classes variations.

L331: which depth? What age?

L358: Provide the age of this H2 tephra

L367: why in the catchment

L379-381: no grain size data presented on this core.

Figure 8: please add the age distribution on this figure

L404: how was define this therehold?

Figure 9: from where these faults are coming? What are the main movement please add this information on the study site part. Please add the Fjord Aysen catchment on this figure to better estimate if it was affected or not. This figure is truly hard to read probability line in white are nit visible.

L457: what is the distance from this site?

References:

Jacq, K., Debret, M., Fanget, B., Coquin, D., Sabatier, P., Pignol, C., Arnaud, F., and Perrette, Y.: Theoretical Principles and Perspectives of Hyperspectral Imaging Applied to Sediment Core Analysis, Quaternary, 5, 28, https://doi.org/10.3390/quat5020028, 2022.

Piret, L., Bertrand, S., Hawkings, J., Kylander, M. E., Torrejón, F., Amann, B., and Wadham, J.: High-resolution fjord sediment record of a receding glacier with growing intermediate proglacial lake (Steffen Fjord, Chilean Patagonia), Earth Surf. Process. Landforms, 46, 239–251, https://doi.org/10.1002/esp.5015, 2021.

Rapuc, W., Sabatier, P., Andrič, M., Crouzet, C., Arnaud, F., Chapron, E., Šmuc, A., Develle, A., Wilhelm, B., Demory, F., Reyss, J., Régnier, E., Daut, G., and Von Grafenstein, U.: 6600 years of earthquake record in the Julian Alps (Lake Bohinj, Slovenia), Sedimentology, 65, 1777–1799, https://doi.org/10.1111/sed.12446, 2018.

Trachsel, M., Grosjean, M., Schnyder, D., Kamenik, C., and Rein, B.: Scanning reflectance spectroscopy (380–730 nm): a novel method for quantitative high-resolution climate reconstructions from minerogenic lake sediments, J Paleolimnol, 44, 979–994, https://doi.org/10.1007/s10933-010-9468-7, 2010.

Vandekerkhove, E., Bertrand, S., Torrejón, F., Kylander, M. E., Reid, B., and Saunders, K. M.: Signature of modern glacial lake outburst floods in fjord sediments (Baker River, southern Chile), Sedimentology, 68, 2798–2819, https://doi.org/10.1111/sed.12874, 2021.

Wilhelm, B., Nomade, J., Crouzet, C., Litty, C., Sabatier, P., Belle, S., Rolland, Y., Revel, M., Courboulex, F., Arnaud, F., and Anselmetti, F. S.: Quantified sensitivity of small lake sediments to record historic earthquakes: Implications for paleoseismology: LAKE SENSITIVITY TO RECORD EARTHQUAKES, Journal of Geophysical Research: Earth Surface, 121, 2–16, https://doi.org/10.1002/2015JF003644, 2016.

---

## Author Response (AR2)

**Date:** 12/07/2024

**Manuscript number:** EGUSPHERE-2024-8

**Title of article:** Co- and postseismic subaquatic evidence for prehistoric fault activity near Coyhaique, Aysén Region, Chile

**Name of Corresponding Author:** Morgan Vervoort

**Email Adress of the Corresponding Author:** Morgan.Vervoort@UGent.be

**RC3:** Anonymous

**RC3 review:** Indicated in blue

Stepping late into this review process, I think the manuscript is interesting and the authors considered earlier referee comments up to a certain degree in their revision. The combination of ground motion modelling and sedimentological analyses to estimate seismic hatard is interesting and potentially worth to be published in NHESS. However, the paper is not up to standards considering the description of the methodological approaches to estimate earthquake ground motion intensities as neither the data used nor the methods applied are shown. This should be fixed in the next round of revision.

Thank you for your review. We carefully went over all the separate comments and made the necessary changes to improve the manuscript. The methodology applied for this paper is developed by Vanneste et al. (2018). A reference to this paper is added in the text, but describing the full methodology (comprising a whole paper on its own) would overshadow the message from this paper. Nevertheless, we added a few additional lines for clarification of the rationale behind the methodology. Additionally, all necessary information (parameters etc.) is available in the text to exactly reproduce the outcomes of this work.

P1L13: Please add abbreviation LOFZ because this is frequently used in the text below
This has been added in the revised manuscript.

Figure 1: I wonder why the authors do not use structural geological maps, maybe together with their relief maps here since this would be much more instructive for a general understanding of the study area tectonics, in combination with a symbolization of the general senses of movement of the major faults plotted.
We changed Figure 1a to a structural geological map. Figure 1b is changed to a zoom of the Aysén Fjord area, with its catchment zone indicated. Figure 1c is the previous Figure 1b, thus a zoom of the Lago Pollux area. We hope that by adding the structural geological map, the general understanding of tectonics will be easier.

P5L145: Please show the location of the samples in Fig. 1. What is Section IX?
The location of the cores are indicated on Figure 1 through a red dot (further indicated in the legend of Figure 1) and is also written in the caption of the figure. On P4L133-134 is stated "The present study focusses on sections VIII and IX of the core (9.5-12 m depth)" with a reference to the supplementary information where the whole core is depicted in Figure S3. However, we added "Section IX of core MD07-3117" on this line to avoid any confusion.

Section 3.3: The ground motion modelling approach described here cannot easily be followed. It would be very helpful to show (at least in the appendix) some earthquake data (at least for strong

motion events) and maybe epicenter- and possibly intensity maps to illustrate the modelling approach.

See the explanation after the general review. Additionally, all epicenter data are theoretical as explained in the methodology, either comprising any location in a grid the surrounding of the two considered lakes, or one of the assumed faults. For ease of understanding, we did add the intensity distribution maps for the theoretical earthquake we consider most likely to have caused our observed pattern of sedimentary shaking evidence.

Figure 2: In A, I would suggest to either plot the bathymetric information into Figure 1B or plot the (tentative?) trace of the "Castor Fault" here. The fault zone should be well observable in Figure 2C but there is no line drawing or annotation? It would be nice to have a discussion on the subvertical structural features in the text. I would also recommend to plot depth-migrated seismic sections. In the appendix, a dense array of seismic scan lines is plotted so I wonder why the structural configuration cannot be better documented?

We chose to not plot the fault traces on the bathymetric map as we found no evidence in the seismic profile of this fault. This is noted on P18L487-488. Since we only have 2D seismic data, depth-migrated seismic sections can not be plotted.

P13L378: Please show some sedimentological evidence why the deposits associated with MTD in unit 5.5 can be interpreted as turbidites (the cited Figures are missing this information).

Since we have no (long) sediment core in Lago Pollux, we are not able to provide sedimentological evidence. However, as stated on P9L267-268, a thin transparent facies with ponding geometry was identified on top of horizon 5.5. We interpret these as turbidite deposits, and references are now added to other studies around the world where ponded units above landslide-events are also interpreted as turbidite deposits.

Figure 7: The maps in B and C have no scale. What are the hatched areas (missing in legend)?

The scales are now added on Figure 7b and c. In the figure caption, it was already stated that the hatched areas represent MTDs.

Figure 9: Again, the earthquake modelling approach and the data used for it is not explained or shown. Please improve; a reference to the geological map of Chile from Sernageomin is not sufficient here. Please also mention in the Caption that the location of the maps is plotted in Figure 2.

See the explanation after the general review and the comment on section 3.3. We now mentioned a reference to Figure 1a for the maps, and the reference to Sernageomin merely comprises the position of the fault traces.

Figure 10: The results of this probabilistic modelling are quite interesting, however due to the missing presentation of data and methods they cannot be followed by the reader. Please improve.

See the explanation after the general review and the comment on section 3.3.

P17L459: Considering the discussion on landslide-affected areas, I wonder if there is some landslide inventory available in this area which may be exploited?

We have not found any records of a landslide inventory in this area.